# DnaJC7 binds natively folded structural elements in tau to inhibit amyloid formation

Zhiqiang Hou [1,2], Pawel M. Wydorski [1,3], Valerie A. Perez [1,3], Aydé Mendoza-Oliva[1], Bryan D. Ryder [1,3], Hilda Mirbaha[1], Omar Kashmer[1] & Lukasz A. Joachimiak [1,4 ✉]

Molecular chaperones, including Hsp70/J-domain protein (JDP) families, play central roles in binding substrates to prevent their aggregation. How JDPs select different conformations of substrates remains poorly understood. Here, we report an interaction between the JDP DnaJC7 and tau that efficiently suppresses tau aggregation in vitro and in cells. DnaJC7 binds preferentially to natively folded wild-type tau, but disease-associated mutants in tau reduce chaperone binding affinity. We identify that DnaJC7 uses a single TPR domain to recognize a β-turn structural element in tau that contains the [275]VQIINK[280] amyloid motif. Wild-type tau, but not mutant, β-turn structural elements can block full-length tau binding to DnaJC7. These data suggest DnaJC7 preferentially binds and stabilizes natively folded conformations of tau to prevent tau conversion into amyloids. Our work identifies a novel mechanism of tau aggregation regulation that can be exploited as both a diagnostic and a therapeutic intervention.

[1] Center for Alzheimer's and Neurodegenerative Diseases, University of Texas Southwestern Medical Center, Dallas, TX, United States. [2] Department of Biophysics, University of Texas Southwestern Medical Center, Dallas, TX, United States. [3] Molecular Biophysics Program, University of Texas Southwestern Medical Center, Dallas, TX, United States. [4] Department of Biochemistry, University of Texas Southwestern Medical Center, Dallas, TX, United States. ✉email: Lukasz.Joachimiak@UTSouthwestern.edu

Molecular chaperones maintain cellular homeostasis by reducing aggregation, preventing misfolding, and stabilizing natively folded conformations. One of the most ubiquitous and conserved chaperone systems is encoded by the 70-kDa heat-shock and the J-domain proteins (Hsp70s/JDPs), which participate in diverse cellular functions that span protein folding, refolding, and degradation[1]. Hsp70/JDP function is linked to regulation of pathogenic protein conformations of tau, α-synuclein, and other proteins that cause neurodegenerative diseases[2–7]. The Hsp70s clearly have an impact on misfolding diseases, but they also play central roles in folding a large portion of the proteome and their dysregulation may have broader impacts[8]. While the Hsp70 protein family is conserved, substrate specificity and selectivity are imparted by the JDP chaperones. JDPs are thought to capture substrates and transfer them to an Hsp70 for subsequent refolding[9]. Coordination of Hsp70 and JDP activities enables high selectivity under diverse cellular functions and broad sequence diversity of substrate proteins, but it is now recognized that JDPs themselves encode the capacity to effectively suppress protein aggregation[10,11].

The human genome encodes 49 JDPs, of which a majority are expressed in the brain[12]. All the members of the family contain a conserved J domain (JD), which binds and activates Hsp70 for substrate transfer. JDPs are classified into A, B, and C subfamilies according to their domain organization. Class-A and -B JDPs are considered general chaperones that bind to nascent, unfolded, or misfolded aggregation-prone substrates and transfer them to Hsp70s[13,14]. Little is known about the capacity of Class-C JDP family members in substrate binding and folding due to their broad diversity of domain organization and likely functions[15]. One such member of the Class-C JDP family, DnaJC7 (TPR2; tetratricopeptide repeat 2 protein), is highly expressed in the brain[16] and loss of DnaJC7 function is implicated in amyotrophic lateral sclerosis (ALS) and other neurodegenerative diseases[17]. DnaJC7, together with Hsp90 and Hsp70, has been implicated in the maturation of the glucocorticoid and progesterone receptors[18,19]. Further, DnaJC7 also plays a role in retaining the constitutive active/androstane receptor (CAR) transcription factor in the cytosol and thus regulating gene activation by CAR[20]. DnaJC7 is distinct among the JDPs in that it contains three tetratricopeptide-repeat (TPR) domains termed TPR1, TPR2a, and TPR2b. Chaperones that contain TPRs comprise approximately one-third of chaperone proteins, but the role of TPRs in substrate binding remains unknown[21]. Another example of a TPR-containing chaperone is the Hsp70–Hsp90-organizing protein (HOP), known as stress inducible protein (STI1) in yeast, which is composed of multiple TPRs that bind to Hsp70, Hsp90, and substrate simultaneously to facilitate substrate transfer between Hsp70 and Hsp90[22]. In addition to TPR domains, DnaJC7 also encodes a J domain, suggesting that if it binds substrates, it may be able to transfer them to Hsp70 like the canonical JDPs.

Here we report that DnaJC7 directly influences the aggregation properties of the microtubule-associated protein tau. Tau is an intrinsically disordered protein that plays important roles in stabilizing microtubules[23]. In its native nondisease conformation, tau is very stable and does not readily aggregate[24]. Recent data suggest that the native conformation of tau monomer can convert into a pathogenic seed that initiates the disease process to amyloid fibril formation linked to many neurodegenerative diseases[25]. We find that DnaJC7 preferentially associates with native conformations of tau isolated from tauopathy mouse brains and CRISPR/Cas9 knockout of DnaJC7 promotes tau aggregation in cells. We reconstituted the DnaJC7–tau interaction in vitro and used a combination of cross-linking–mass spectrometry (XL–MS) and NMR to identify the binding surfaces between DnaJC7 and

tau. DnaJC7 uses the TPR2b domain to interact with a sequence element in the repeat domain of tau located between repeats 1 and 2, termed R1R2. We show differences in affinity between WT tau and an aggregation-prone disease-associated P301L tau mutant that can be explained by changes in the tau conformation, suggesting that DnaJC7 preferentially binds to the natively folded conformations of wild-type tau. In vitro and in cells, DnaJC7 binding to tau efficiently suppresses seeded tau aggregation. While each TPR domain individually can bind tightly to tau, the intact DnaJC7 chaperone is essential for aggregation-suppression activity in vitro. We employ XL–MS guided modeling to show how DnaJC7 binds to R1R2 in a natively folded conformation consistent with NMR chemical shifts. Mutations in the peptide-binding grove in TPR2b of DnaJC7 reduced tau-binding affinity, rendering the mutant less efficient at suppressing tau aggregation. Finally, we show that an isolated natively folded R1R2 minimal fragment derived from tau can block DnaJC7 binding to tau, while an R1R2 peptide encoding disease-associated proline to serine mutation cannot. Overall, our findings indicate that DnaJC7 binds to natively folded tau, is an important modulator of tau aggregation in vitro and in cells, and may play a role in preventing the formation of pathogenic seeds in disease.

## Results

**DnaJC7 influences tau seeding in vivo and in cells.** We have recently shown that tau seeding species appear in a tauopathy mouse model (PS19) prior to the appearance of tau pathology in the brain[26]. We sought to identify proteins that are bound to small soluble tau species in this early window when the first soluble seeding species appear to gain insight into cellular factors that could play a role in influencing tau aggregation. Soluble tau was immunoprecipitated (IP) from young PS19 mouse brains aged from one week to six weeks and small species were isolated using size exclusion chromatography (SEC)[26]. Each sample was assayed for seeding capacity determined by transducing samples into tau biosensor cells expressing tau-repeat domain (herein tauRD) fused to CFP and YFP[27] and the extent of seeding was quantified by measuring the FRET signal between the fluorescent fusion constructs, which reports on their intracellular assembly into amyloids (Fig. 1a). In parallel, we employed mass spectrometry to identify cofactors bound to the immunoprecipitated tau. We identified many known tau interactors, including Hsp70s, 14-3-3 s, prolyl isomerases, Hsp90, and a series of JDPs (Source Data 1). Overall, we discovered nine JDPs that copurified with tau across each tau IP (Source Data 1). Among them, we identified JDPs, including DnaJA2, DnaJB1, and DnaJB6, which have been previously implicated in binding amyloidogenic substrates to control fibril assembly or disassembly[28–30]. We find that tau signal intensity increased ten-fold across this time window and most chaperones trend with tau signal with some notable exceptions: signal intensity for DnaJC7 decreased five-fold, while for DnaJC5, it increased nearly 50-fold (Supplementary Fig. 1a). To understand the abundance of the different JDPs bound as a function of tau changes, we normalized the JDP signal intensity according to the 1N4R tau intensities (see "Methods"). Recently, JDPs have been implicated in directly modulating protein aggregation[28–30]. We wanted to determine how the levels of these associated JDPs changed as a function of tau seeding activity quantified in tau biosensor cells. Many of the JDPs, including DnaJA1, DnaJC6, DnaJA2, DnaJB4, and DnaJA4, were associated at different levels, but the normalized signal remained flat as tau seeding increased with mouse age (Supplementary Fig. 1b). The normalized abundance for DnaJC5 increased ten-fold over this window (Supplementary Fig. 1b). For DnaJB6 and DnaJB1, we observed a two-fold and three-fold decrease in normalized

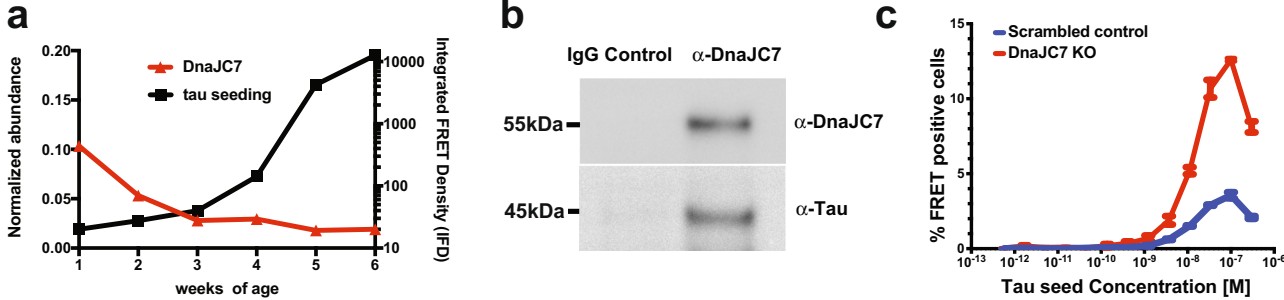

**Fig. 1 Identification of a JDP DnaJC7 that binds to tau and influences seeding. a** Immunoprecipitation of tau from PS19 tauopathy mice across different ages (weeks 1 through 6) reveals an increase in seeding activity at early ages (black), which anti-correlates with a five-fold decrease of JDP DnaJC7 abundance (red). Isolated tau from the different samples was evaluated for the presence of pathogenic seeds using tau biosensors (right y-axis) and the normalized abundance of DnaJC7 (relative to P301S 1N4R tau; [$I_{JDP}/I_{tau}$]*100) was determined using mass spectrometry (left y axis). Seeding experiments were performed as biological triplicates and the mass spectrometry data were analyzed using Proteome Discoverer (Thermo). Each time point (weeks 1 through 6) involved three mouse brains ($n = 3$). **b**. Co-immunoprecipitation of DnaJC7 and tau from PS19 mouse brain lysates. Brain lysates were immunoprecipitated with an anti-DnaJC7 antibody or an IgG isotype antibody as negative control. The immunoprecipitates were analyzed by western blot analysis and probed with anti-DnaJC7 or anti-tau antibodies. IP and western blot was carried out 3 independent times. **c**. Dose titration of recombinant heparin-induced tau fibril seeding on the tauRD mClover3/mCerulean biosensor cells. Cells were treated with tau fibril concentrations performed as three-fold dilutions ranging from 300 nM down to 565 fM. Seeding was quantified via flow cytometry based on the percentage of FRET-positive cells in each sample. The decrease in seeding at 300 nM corresponds to cell death due to toxicity of high concentrations of tau fibrils. Seeding experiments were performed as biological replicates ($n = 3$) and are shown as a mean with standard deviation.

abundance, respectively (Supplementary Fig. 1b), while for DnaJC7, we found a five-fold decrease in its normalized abundance as tau seeding increased (Fig. 1a and Supplementary Fig. 1b). These data indicate that while many JDPs can bind tau independent of seeding capacity, some may have selectivity for different conformations of tau and can discriminate between native vs aggregation-prone conformations. We were intrigued by the behavior of DnaJC7, which is not known to bind substrates nor has it been implicated directly in protein-aggregation processes. We first tested whether DnaJC7 binding to tau can be directly detected by western blot following IP from PS19 mouse brain tissues expressing human 1N4R tau[31]. DnaJC7 was immunoprecipitated from PS19 mouse brain lysates using an anti-DnaJC7 antibody or an IgG isotype as a negative control. Immunoblotting revealed that tau co-immunoprecipitates with DnaJC7, while no signal was observed in the negative control (Fig. 1b and Supplementary Fig. 1c).

We then carried out a DnaJC7 knockout experiment in tau biosensor cells to determine if the loss of the tau:DnaJC7 interaction could change tau seeding activity in tau biosensor cells. Four CRISPR guide RNAs (gRNAs) targeting DnaJC7 (Supplementary Table 1) were used as a cassette to disrupt the DnaJC7 gene in the HEK293 tau biosensor line[32]. In parallel, a nontargeting gRNA sequence was used as a negative control. Western blot of cell lysates was used to confirm the knockout of DnaJC7 (Supplementary Fig. 1d). Compared with the negative control, we observed a noteworthy four-fold increase in tau seeding capacity in DnaJC7 knockout cells induced by tau fibrils (Fig. 1c). Our ex vivo tau:DnaJC7-binding data and DnaJC7 knockout effects on tau seeding in cells suggest that DnaJC7 could play a crucial role in preventing tau aggregation in vivo.

**DnaJC7 binds tauRD through the R1R2 interrepeat element**. We first employed binding assays and structural approaches to determine how DnaJC7 directly interacts with tauRD to influence its aggregation (Fig. 2a). DnaJC7 encodes three TPR domains (TPR1, TPR2a, and TPR2b) followed by a C-terminal J domain (Fig. 2a). The 34-residue TPR motif is composed of a two-helix bundle. Tandem TPR motifs form a ubiquitous protein interaction module that binds to peptides via a canonical binding

groove. We used a combination of microscale thermophoresis (MST)-binding measurements, solution NMR, and cross-linking mass spectrometry (XL-MS) to probe how DnaJC7 interacts with wild-type tauRD (WT tauRD) and a frontotemporal dementia disease-associated P301L mutant (P301L tauRD) (Fig. 2a). To investigate whether DnaJC7 directly binds to tauRD, we used MST to measure binding affinity between DnaJC7 and WT tauRD or P301L tauRD derivatized with a cyanine dye (Cy5–NHS ester) (Fig. 2b and Supplementary Fig. 2a). Our measurements revealed that DnaJC7 binds to WT tauRD with $2.2 \pm 0.4\ \mu M$ binding affinity, while P301L tauRD bound with a three-fold lower-binding affinity relative to WT tauRD ($6.2 \pm 1.4\ \mu M$). The P301L mutation is located in a conserved β-hairpin motif just upstream of the 306VQIVYK311 amyloid motif[33]. Prior work from our group has shown that this mutation preferentially unfolds this structural motif to drive exposure of 306VQIVYK311 that promotes aggregation[34]. While other chaperones, including Hsc70/Hsp70, Hsp90, and JDPs, have been shown to bind to 306VQIVYK311 to suppress tau aggregation[29,35], it is surprising that unfolding of this motif in the mutant decreases affinity for DnaJC7.

To probe the domain interactions between DnaJC7 and tauRD in more detail, we employed XL-MS to map intermolecular sites of interaction. XL-MS provides direct binary-interaction pairs between residues within a structure while also identifying binding contacts in protein complexes in a resolution range of 7–30 Å[36]. The preformed complexes were reacted with disuccinimidyl suberate (DSS), quenched, and heterodimer bands corresponding to the complex were excised from an SDS-PAGE gel. The samples were processed to isolate cross-linked peptides (see "Methods") and the peptides were analyzed by mass spectrometry to identify the cross-link pairs using our established XL-MS pipeline (Source Data 2)[37]. We only considered cross-link pairs with high score cutoffs that occurred across replicate samples. We identified four cross-links between K280, K281, and K311 of WT tauRD and K254 and K306 of DnaJC7 (Fig. 2c and Supplementary Fig. 2b). The tauRD cross-links localize predominantly to the interrepeat regions that span the repeat 1 and 2 interfaces (herein R1R2) and include the 275VQIINK280 amyloid motif (Fig. 2c and Supplementary Fig. 2b), with one cross-link also localizing to the repeat

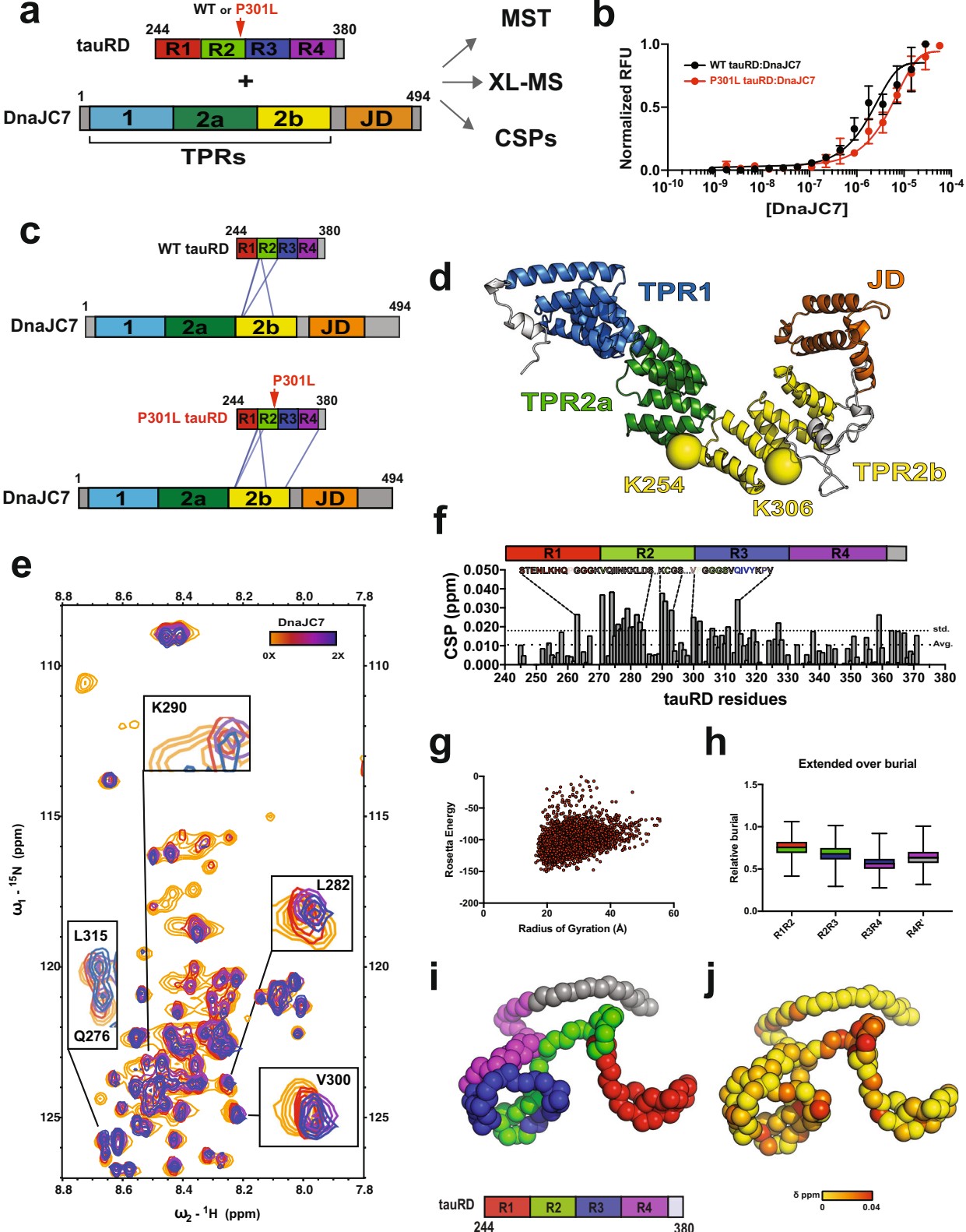

2 and 3 interfaces (herein R2R3) in proximity to the [306]VQI-VYK[311] amyloid motif. Mapping these cross-links onto a homology model of DnaJC7, built using Rosetta[38], reveals that the K254 and K306 cross-links are located in the peptide-binding groove in TPR2b (Fig. 2d; yellow spheres). While the C-terminal JD and TPR2b domains encode the largest number of lysine residues per domain, we only observe cross-links to TPR2b

(Supplementary Fig. 2c). Furthermore, the N-terminal TPR1 and TPR2a domains each contain 9 and 11 lysines, respectively, but do not cross-link to the substrate (Supplementary Fig. 2c), indicating that binding and not frequency of lysines determines cross-links. Nearly identical cross-link patterns were identified between P301L tauRD:DnaJC7 (Fig. 2c). Like the homologous

**Fig. 2 DnaJC7 binds directly to tauRD through the R1R2 inter-repeat element. a** Schematic of the DnaJC7 and tauRD constructs used. TauRD domains are colored in red, green, blue, and magenta for repeats 1, 2, 3, and 4, respectively. DnaJC7 domains are colored in light blue, dark green, yellow, and orange for TPR1, TPR2a, TPR2b, and J domain, respectively. **b** MST assay to quantify affinity between tauRD:DnaJC7 (black) and P301L tauRD:DnaJC7 (red). The MST-binding experiments were performed as technical replicates ($n = 3$) and each concentration is shown as a mean with standard deviation. The data were fit to a linear-regression model to estimate the binding constant. **c** XL-MS to identify the interactions in the complex. Each protein is shown as cartoon and the cross-links are shown as blue lines linking the two cross-linked positions. The constructs are colored as in (**a**). **d** DnaJC7 homology model shown in cartoon representation and the domains are colored as in (**a**). The two major cross-link sites in TPR2b are shown as spheres and colored by the domain. **e** 2D $^1$H-$^{15}$N HSQC spectra of tauRD in the absence (red) and in the presence of increasing DnaJC7 concentrations. Molar ratio of tauRD to DnaJC7 is 2:1 (yellow), 1:1 (red), and 1:2 (blue). Peaks that are perturbed in the presence of DnaJC7 are shown as insets and labeled by the amino acid position. **f** NMR CSPs of tauRD in a 1:1 molar ratio of DnaJC7 are shown as a bar plot colored in gray. The amino acid residues for tauRD that are shifted are shown above the barplot, the remaining residues are colored as in (**a**). The tauRD cartoon is shown above the plot and the domains are colored as in (**a**). Average and standard deviations are shown as dashed lines. **g** Rosetta energy and calculated radius of gyration is shown for an ensemble of tauRD models built by CS-ROSETTA. **h** Ensemble average of normalized solvent exposure (normalized to solvent exposure of a fully extended chain) for inter-repeat elements R1R2, R2R3, R3R4, and R4R′ is shown as a boxplot and colored red/green, green/blue, blue/magenta, and magenta/gray, respectively. Solvent-exposure statistics for the tauRD sequence elements were calculated from $n = 5000$ structural models. The box-and-whisker plots are shown with minima, maxima, and center, plotted at the 25$^{th}$ percentile. **i** Representative model of tau RD from the ensemble. The backbone of the model is shown in spacefill and is colored as in (**a**). **j** The backbone of the model is shown in spacefill. tauRD model is colored according to DnaJC7-induced CSPs color-coded from yellow (0 ppm) to red (0.04 ppm).

---

$^{306}$VQIVYK$^{311}$, $^{275}$VQIINK$^{280}$ is also important for tau assembly and fibrillization[33] and chaperones, including DnaJA2 and Hsp70, have been shown to recognize these two amyloid motifs[29]. Our data support that DnaJC7 interacts with tau via a defined TPR-based interaction, utilizing a peptide-binding groove to recognize elements that include the amyloid motifs.

To validate the tauRD region involved in binding to DnaJC7, we acquired 2D $^1$H-$^{15}$N HSQC spectra of $^{15}$N tauRD alone and titrated with DnaJC7. TauRD assignments were transferred from previously published data[39] (Supplementary Fig. 2d). The addition of DnaJC7 to the $^{15}$N tauRD revealed concentration-dependent chemical shifts as highlighted by changes at Q276, L282, and K290 (Fig. 2e, inset). Systematic plotting of all chemical shift perturbations (CSPs) revealed changes that localized to the R1R2 element extending through R2 and were consistent with the XL-MS-site identification (Fig. 2f). In addition to CSPs, we also observe broadening of peaks in the R1R2 element, including S258, S262, T263, N265, and K274 (Supplementary Fig. 2e). Additionally, we find that degenerate glycine peaks, including G271/G302, that localize to the P-G-G-G motifs preferentially broaden, suggesting that DnaJC7 may directly recognize the turn motif in the R1R2 element (Supplementary Fig. 2e). Together, our NMR data suggest that DnaJC7 binds to the R1R2 element, including the P-G-G-G turn, using a combination of slow and fast-exchange kinetics. Protein–protein interactions generally only occur on nonburied surface; as such, we speculate that the identified interrepeat elements are likely more solvent-exposed. To identify solvent-exposed regions in tauRD, we employed CS-Rosetta[40] guided by tauRD chemical shifts to produce an ensemble of conformations that are consistent with SAXS-derived sizes (Fig. 2g). By calculating the solvent exposure of each interface element in the ensemble and normalizing to the unfolded state, we revealed that the R1R2 element is on average more solvent-exposed (Fig. 2h, colored in red/green). We show a representative model of tauRD colored by the repeat domain to highlight the preferential exposure of the R1R2 elements (Fig. 2i, interface of red/green), which are compatible with NMR-based CSPs on tauRD in response to DnaJC7 binding (Fig. 2j).

**Pathogenic mutations in tau modify its conformation and recognition by DnaJC7.** While our initial data support that DnaJC7 can bind and modify aggregation of tauRD, our IP data support that DnaJC7 can recognize full-length tau and may have binding preference for natively folded conformations of tau. To

gain insight into how DnaJC7 binds to full-length 2N4R WT and P301L tau, the binding mode of interaction between tau and the chaperone, and possible changes in tau conformation in response to pathogenic mutations, we again employed binding assays and XL-MS. For the cross-linking analysis, we first interpreted intramolecular cross-links for WT and P301L tau, followed by analysis of contacts derived from the WT tau:DnaJC7 and P301L tau:DnaJC7 complexes.

Using MST, we determined that DnaJC7 binds to full-length tau with $471 \pm 53$ nM affinity (Fig. 3a and Supplementary Fig. 3a), nearly five-fold tighter than WT tauRD. We speculated that the N-terminus (N-term, residues 1–243) could independently contribute to the binding affinity, but binding measurements for the N-terminal fragment alone revealed an affinity of $17 \pm 4$ μM (Supplementary Fig. 3a). These data hinted at possible cooperativity between the N-terminus and the repeat domain. To further corroborate this observation, we measured the affinity for the full-length P301L tau mutant and DnaJC7 and determined the affinity to be $8.3 \pm 2.2$ μM, which is nearly 20-fold lower than WT (Fig. 3a and Supplementary Fig. 3a). Taken together, DnaJC7 binds full-length tau with nanomolar affinity and pathogenic mutations in full-length tau decrease affinity to levels similar to WT and P301L tauRD. Consistent with this idea, we also measured binding of DnaJC7 for a recombinant tau monomer seed[41] and found that it binds weakly with $15.5 \pm 2.2$ μM affinity (Supplementary Fig. 3a). These data suggest that DnaJC7 preferentially binds to a natively folded conformation of full-length tau with high affinity in which the N-terminus and the repeat domain are more stably associated. We hypothesize that the P301L tau mutation shifts the conformation toward an aggregation-prone seeding conformation that has lower affinity for DnaJC7.

To test this idea, we first employed XL-MS to probe how conformational changes in full-length WT and P301L tau could dictate the observed differences in affinity for DnaJC7. WT and P301L tau were reacted with DSS or the zero length cross-linker 4-(4,6-dimethoxy-1,3,5-triazin-2-yl)-4-methylmorpholinium (DMTMM) at 25 °C and 50 °C, quenched, and the cross-linked monomer band was extracted from SDS-PAGE gels (Supplementary Fig. 3b). We employed these two temperatures to probe the stability of the interactions in response to unfolding. The cross-links were processed through our XL-MS pipeline as replicates yielding similar spectral intensities across replicates (Source Data 2) and only high-scoring pairs present across replicate datasets were considered (Source Data 2). We identified 103 and 41

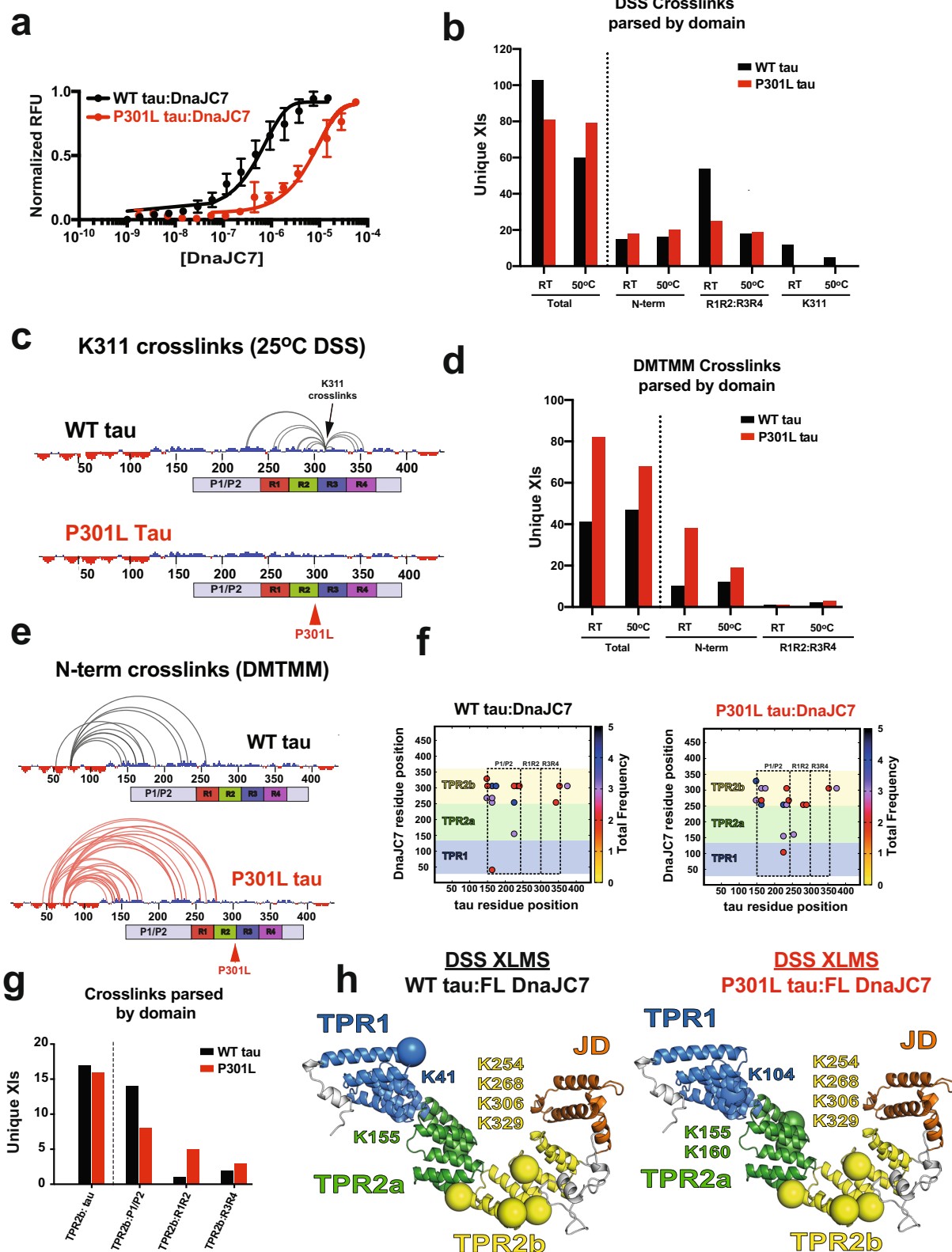

consensus cross-links at 25 °C for WT tau using the DSS and DMTMM chemistries, respectively. The reactions performed at 50 °C for WT tau yielded a nearly two-fold drop in DSS cross-links (Fig. 3b, 60 pairs), while the number of DMTMM pairs remained flat at 47 cross-links (Fig. 3d). For P301L tau, we identified 81 and 82 consensus cross-links using the DSS and DMTMM chemistries at 25 °C, respectively. In contrast to the

WT tau datasets, at 50 °C, the numbers of consensus cross-links for P301L tau remained relatively flat with 79 and 68 for the DSS and DMTMM chemistries, respectively. Each cross-linking chemistry captures different types of interactions; DSS traps contacts between primary amines (i.e., lysines), while DMTMM traps contacts between primary amines (i.e., lysines) and carboxylic acids (i.e., aspartic and glutamic acids). We surmised

**Fig. 3 Pathogenic P301L mutation alters tau conformation and impacts DnaJC7-binding affinity. a** MST-binding assay for full-length WT tau:DnaJC7 (black) and P301L tau:DnaJC7 (red). The MST-binding experiments were performed as technical replicates ($n = 3$) and are shown as a mean with standard deviation. The data were fit to a linear-regression model to estimate the binding constant. **b** Unique consensus DSS cross-link pairs identified in WT (black) and P301L (red) tau are shown as a barplot comparing cross-links for total parsed by different domains (N-terminus, R1R2:R3R4 and K311) and acquired at RT and 50 °C. **c** Consensus cross-links involving K311 are shown for WT and P301L tau. The cross-links are shown as semi-circles mapped onto the linear sequence of tau. The net charge per residue (NCPR) distribution is shown on the sequence axis and is colored red and blue for acidic and basic residues, respectively. Cartoon domain for tau is shown below the sequence axis, the domains are colored as in Fig. 2a. **d** Unique consensus DMTMM cross-link pairs identified in WT (black) and P301L (red) tau are shown as a bar plot comparing total cross-links and cross-links parsed by different domains (N-terminus and R1R2:R3R4) and acquired at RT and 50 °C. **e** Consensus cross-links between the acidic N-terminus and the basic regions of tau (mostly repeat domain) for WT (black) and P301L tau (red). NCPR distribution is shown as in (**c**). Cartoon domain for tau is shown below the sequence axis, the domains are colored as in Fig. 2a. **f** Consensus WT tau:DnaJC7 (top) and P301L tau:DnaJC7 (bottom) intermolecular DSS cross-link contact map colored by average frequency across replicates. **g** Unique consensus DSS cross-link pairs identified in WT tau:DnaJC7 (black) and P301L tau:DnaJC7 (red) are shown as a barplot comparing total cross-links and cross-links parsed by TPR2b of DnaJC7 contacts to different regions on tau (N-terminus, R1R2 and R3R4). **h** The major cross-link sites derived from the WT tau:DnaJC7 and P301L tau:DnaJC7 XL-MS experiments. Four cross-links were detected to TPR2b in each experiment (K254, K268, K306, and K329). We additionally detect a single-consensus cross-link to TPR1 (K41) and TPR2a (K155) in the WT tau:DnaJC7 complex. In the P301L tau:DnaJC7 complex, we also detect a consensus cross-link to TPR1 (K104) and two cross-links to TPR2a (K155 and K160). DnaJC7 homology model shown in cartoon representation with the domains colored as in (Fig. 2). The cross-link sites are shown as spheres and colored by the domain.

that DSS would predominantly capture contacts within the basic repeat domain, while the DMTMM chemistry would trap interactions between the acidic termini and the basic repeat domain. Indeed, DSS captures more contacts within the repeat domain compared with the N-terminus for both WT and P301L tau (Fig. 3b and Supplementary Fig. 3c). We also observed that in WT tau, we captured many more DSS cross-links in the repeat domain compared with P301L tau at 25 °C. Increasing the temperature reduced the number of repeat-domain cross-links by two-fold in WT and only reduced them by 25% in P301L tau (Fig. 3b). This difference is highlighted when comparing the number of cross-links from the 25 °C experiment in proximity to residue 301. In WT tau, we observe twelve contacts involving K311 that are completely absent in the P301L tau mutant (Fig. 3c). Consistent with domain-charge distributions, in the DMTMM reactions, we detect overall more contacts between the acidic N-terminus and the basic repeat domain and much less within the basic repeat domain (Supplementary Fig. 3d). In P301L tau, we observe nearly 40 cross-links involving the N-term at 25 °C and the number drops nearly two-fold at higher temperature. In contrast, we observe less N-terminal cross-links in WT tau and it appears relatively insensitive to temperature (Fig. 3d). This is further highlighted by the difference in the number of contacts when comparing P301L and WT tau. Taken together, these XL-MS experiments reveal differences in the distribution of contacts within WT and P301L tau underscored by changes in repeat domain in response to pathogenic mutations (Fig. 3c). This is consistent with our previous work on tauRD[34] and expands the conformational changes in full-length tau to also include perturbations of long-range contacts between the acidic N-terminal contacts with the more basic proline-rich and repeat domains.

Our data support that the P301L mutation significantly changes the conformation of the repeat domain in proximity to the mutation and also affects the overall conformation of the acidic N-terminus with respect to the repeat domain. We surmise that changes in tau conformation impact the different affinities between WT and P301L tau toward DnaJC7. It is unclear how conformation of the tau N-terminus alters binding to DnaJC7. We again turned to XL-MS to probe the WT tau:DnaJC7 and P301L tau:DnaJC7 complexes to gain insight into which domains mediate the interaction. As in prior experiments for tauRD:DnaJC7, preformed complexes between WT tau:DnaJC7 and P301L tau:DnaJC7 were reacted with DSS, quenched, and the bands corresponding to the heterodimer complex extracted from SDS-

PAGE gels. The samples were processed and analyzed by LC–MS/MS using our XL-MS pipeline to identify consensus cross-links across replicates with high-score cutoffs (Source Data 2). Similarly to how tauRD binds to DnaJC7, the major binding site that binds WT and P301L tau on the chaperone is TPR2b (Fig. 3f). However, partitioning the identified TPR2b cross-links to tau revealed data consistent with the P301L mutant tau being more unfolded (Fig. 3f). Although the overall numbers of tau cross-links to the TPR2b domain were similar between WT and P301L tau, they appear to partition differently across P1/P2 sites in the proline-rich domain and the repeat domain in tau (Fig. 3g). The TPR2b contacts all localize to the substrate-binding groove of the domain (Fig. 3h) and overlap with contacts detected in DnaJC7 complexed with tauRD (Fig. 2d). In WT tau, we observe nearly five-fold less cross-links between TPR2b and R1R2 and two-fold more cross-links between TPR2b and P1/P2 sites compared with P301L tau. This is because DnaJC7 binds to R1R2 and buries this element in the binding pocket that yields fewer cross-links.

Finally, analysis of tau intramolecular contacts from the XL-MS datasets derived from WT tau:DnaJC7 complexes showed a reduction of long-range contacts with the N-terminus, while in the P301L tau:DnaJC7 complex, these persist, suggesting that DnaJC7 binds preferentially to the folded conformation of WT tau (Supplementary Fig. 3e). Thus, it appears that a more stable interaction between the N-terminus and the repeat domain observed in WT tau preferentially leads to higher-affinity binding, resulting in exposure of the P1/P2 sites and burial of the R1R2 in the tau:DnaJC7-binding interface. Consistent with this observation, both the similarly "unfolded" WT and P301L tauRD have comparable affinity to DnaJC7 as the aggregation-prone full-length P301L tau. To further corroborate our findings, we measured the frequency of total chemical modifications (i.e., monolinks and looplinks) as a proxy for changes in solvent exposure. We found that the P1/P2 region has a six-fold higher rate of modification in WT tau:DnaJC7 compared with WT tau alone (Supplementary Fig. 3f). This region is comparably modified in P301L tau and P301L tau:DnaJC7, but approximately two-fold lower than in WT tau:DnaJC7 (Supplementary Fig. 3f). In contrast, we observed that the modification frequency in the R1R2-interface region dropped two-fold in the WT tau:DnaJC7 complex relative to the tau alone (Supplementary Fig. 3f). These data reaffirm our intermolecular cross-link data on the complexes and support that P301L tau is more unfolded than WT tau. Taken together, these data suggest that the N-terminus plays an

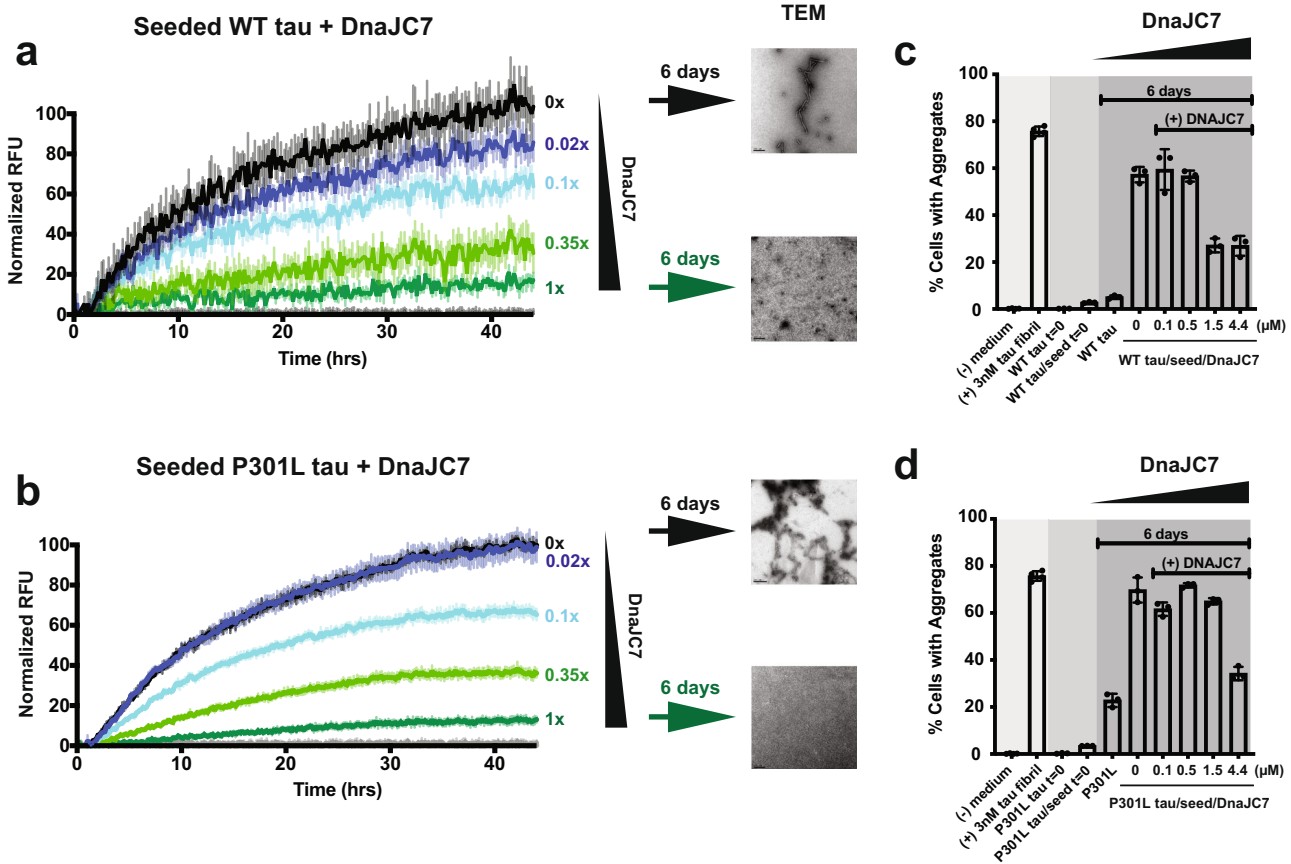

**Fig. 4 DnaJC7 efficiently suppresses tau aggregation in vitro.** 4.4 μM WT tau (**a**) and 4.4 μM P301L tau (**b**) aggregation was induced using 50 nM tau monomer seeds in the presence of different DnaJC7 chaperone concentrations. Aggregation was monitored using ThT fluorescence. Experiments were performed as technical replicates ($n = 3$) and are shown as a mean with standard deviation. DnaJC7 concentrations ranged from 0X (0 μM), 0.02X (0.1 μM), 0.1X (0.5 μM), 0.35X (1.5 μM), to 1X (4.4 μM) and are colored from black to green. Gray curves show negative controls: tau alone, DnaJC7 alone at each concentration, and tau monomer seeds alone. TEM images were taken at the end point to confirm the presence or absence of fibrils. Scale bar for each micrograph is 200 nm. (**c**, **d**) The endpoint and control samples of the in vitro aggregation experiment were assayed for seeding activity in tau biosensor cells. The in-cell aggregation assay shows significant reduction in seeding in the presence of DnaJC7 even after six days of incubation and is consistent with the in vitro measurements. Tau seeding experiments were performed as biological replicates ($n = 3$) and are shown as mean with standard deviation.

important role in defining the topology of the native tau "fold," which is altered by pathogenic mutations. While the tauRD fragment is a good proxy for aggregation, it lacks the regulatory elements that define the native conformation observed in full-length tau.

**DnaJC7 efficiently inhibits aggregation of tau in vitro.** While DnaJC7 appears to impact tau seeding in cells and binds natively-folded tau with submicromolar affinity in vitro, we wanted to test whether DnaJC7 alone can control tau aggregation in vitro. We monitored tau aggregation using a Thioflavin T (ThT) fluorescence aggregation assay to determine the capacity of DnaJC7 to suppress the formation of tau fibrils when tau monomer is induced with substoichiometric amounts of recombinant isolated tau monomer seeds[26]. 4.4 μM wild-type tau monomer was incubated with 50 nM recombinant tau monomer seeds in the presence of different concentrations of DnaJC7, including controls to monitor the behavior of each component (i.e., tau alone, tau monomer seeds alone, and DnaJC7 alone; see "Methods" for details). We found that equimolar concentrations of 4.4 μM DnaJC7 (1:1) yielded efficient suppression of seeded 4.4 μM tau aggregation. Further, we found that even substoichiometric (44:1) amounts of 0.1 μM DnaJC7 reduced ThT signal by 20% over the

time course (Fig. 4a). Consistent with the ThT signal, at the end of the time course, we observed tau fibrils in reactions without DnaJC7 as determined by TEM, while fibrils were not detected in the presence of DnaJC7 (Fig. 4a). Parallel experiments carried out with P301L tau revealed that DnaJC7 has a weaker capacity to suppress seeded 4.4 μM tau aggregation, highlighted by the absence of ThT signal reduction at the 44:1 ratio, which is consistent with lower affinity for this interaction (Fig. 4b).

To confirm the presence of tau aggregates in our in vitro samples, we employed an in-cell tau biosensor assay to detect the seeding capacity of exogenously delivered tau using FRET[27]. Endpoints from our in vitro aggregation experiment (above) were transduced into tau biosensor cells, including negative and positive controls, and measured FRET to quantify the amount of tau aggregates in each sample. Our positive (recombinant tau fibrils) and negative (lipofectamine alone) controls yielded $76 \pm 1\%$ and $0.2 \pm 0.2\%$ FRET signal, respectively. Consistent with the ThT assays, transduction of the in vitro-incubated samples, revealed a reduction in seeding capacity as a function of DnaJC7 concentration (Fig. 4c, d and Supplementary Fig. 4a). In the samples without DnaJC7, we detected $59 \pm 9\%$ of the cells with aggregates and at 3:1 and 1:1 tau:chaperone ratios, the number of cells with aggregates decreased by two-fold over the course of the experiment to $27 \pm 3\%$ and $27 \pm 4\%$, respectively.

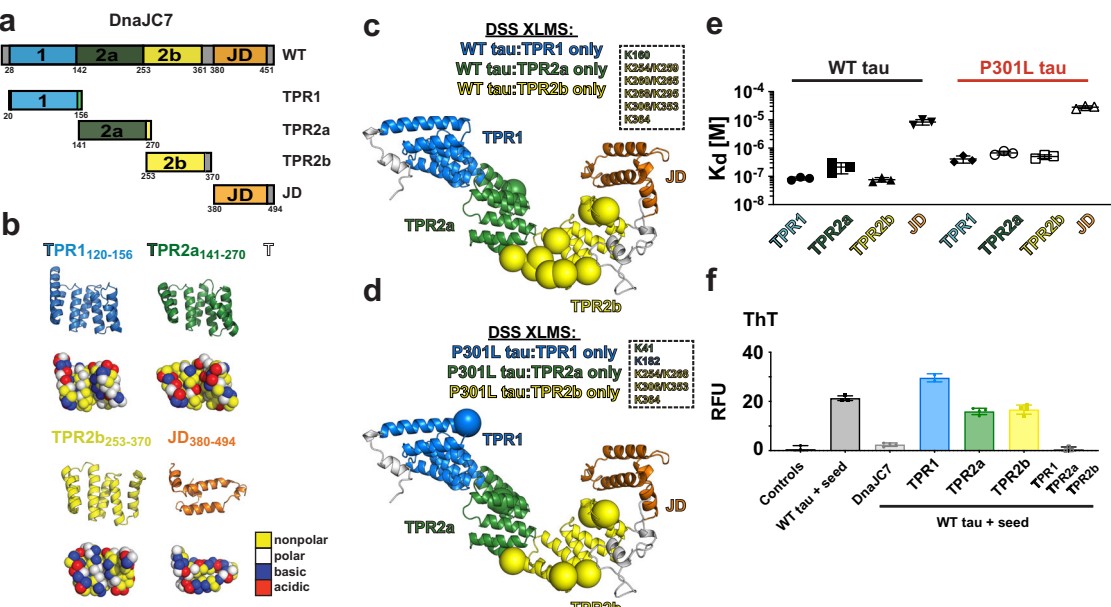

**Fig. 5 Efficient DnaJC7 suppression of tau aggregation relies on all three TPR domains. a** Schematic diagram of DnaJC7 domain constructs used: $TPR1_{20-156}$ (light blue), $TPR2a_{141-270}$ (dark green), $TPR2b_{253-370}$ (yellow), and $JD_{380-494}$ (orange). **b** Cartoon representation of $TPR1_{20-156}$ (light blue), $TPR2a_{141-270}$ (dark green), $TPR2b_{253-370}$ (yellow), and $JD_{380-494}$ (orange) oriented to highlight the peptide binding groove. Spacefill representation of each domain is colored according to amino acid properties; polar (white), nonpolar (yellow), acidic (red), and basic (blue) residues. **c**, **d** The major cross-link sites derived from the WT tau and P301L tau bound to individual TPR domains: TPR1 (blue), TPR2a (green) and TPR2b (yellow). We observe one cross-link from WT tau to TPR2a (green) and nine cross-links to TPR2b (yellow). For P301L tau, we detected one cross-link to TPR1 (blue) and one to TPR2a (green) and four cross-links to TPR2b. DnaJC7 homology model shown in cartoon representation and the domains are colored as in (Fig. 2). The cross-link sites are shown as spheres and colored by the domain. **e** MST-binding affinity summary of TPR construct binding to WT tau and P301L tau. Binding experiments were performed as technical triplicates and fitted to a linear-regression curve to estimate binding constants. The derived binding constants are shown as the mean with standard deviations across replicate experiments ($n = 3$). WT tau and P301L tau binding experiments are shown in black and red, respectively, and the TPR labels are colored according to the TPR domain as in (**a**). **f** Individual TPR domains were used in a ThT fluorescence aggregation assay to measure their effect on suppressing tau aggregation in vitro. The end point of each experiment is shown as a barplot and colored according to the TPR domain in (**a**). The in vitro aggregation experiments were performed as technical replicates ($n = 3$) and the data are reported as a mean with standard deviation.

Parallel seeding experiments with the P301L tau samples across a range of DnaJC7 concentrations revealed nearly $70 \pm 5\%$ of cells contained aggregates in the absence of DnaJC7. To achieve two-fold inhibition of tau seed formation over six days, 1:1 ratios of P301L tau:DnaJC7 were required ($34 \pm 3\%$ of cells), consistent with our finding that DnaJC7 has a lower affinity for P301L tau than WT tau. We also examined the effects of a canonical JDP, DnaJA1, on tau aggregation. DnaJA1 showed only a minor effect of tau aggregation relative to DnaJC7 (Supplementary Fig. 4b). Our data support that DnaJC7 is a potent suppressor of tau aggregation by efficiently binding to natively folded conformations of tau.

**Individual TPR domains bind tightly but do not suppress tau aggregation**. In general, TPR domains, like the three found in DnaJC7, can be seen as scaffolds to mediate protein interactions. Our XL-MS data support that TPR2b in DnaJC7 is the main domain that recognizes tau. To understand how DnaJC7 regulates tau aggregation, we set out to determine whether individual TPR domains ($TPR1_{20-156}$, $TPR2a_{141-270}$, and $TPR2b_{255-370}$) or the J domain$_{380-494}$ can bind and suppress tau aggregation (Fig. 5a). Each individual domain was expressed recombinantly, purified using a two-step purification protocol ("Methods"), and confirmed to be monomeric using SEC, dynamic light scattering and XL-MS. Although TPRs have an overall similar structure, the charge distribution on each domain is different (Fig. 5b). Comparison of the surface properties of the binding grooves on TPR1, TPR2a, and TPR2b reveals that their binding grooves are similar

with some differences (Fig. 5b). Notably, TPR1 and TPR2b encode more basic residues, while TPR2a is more acidic (Fig. 5b).

We first wanted to determine if the isolated TPR domains bind to tau similarly to the full-length DnaJC7. To identify how each individual domain interacts with tau, we again employed XL-MS. Briefly, complexes between the individual DnaJC7 domains and tau were preformed, reacted with DSS, quenched, and the species resolved by SDS-PAGE to extract bands corresponding to the heterodimer (Supplementary Fig. 5a). Samples were processed, analyzed by mass spectrometry, and cross-linked sites were identified using our XL-MS pipeline. The identified cross-links between tau and each single TPR domain imply that the individual domains bind in a largely distributed fashion, suggesting a loss of specificity in tau binding relative to full-length DnaJC7 (Supplementary Fig. 5b). Importantly, the individual TPR-domain cross-link experiments show that they each are capable of binding to tau but the number of observed contacts is dominated by the TPR2b domain (Supplementary Fig. 5b). Interestingly, despite the distributed nature of the TPR2b contacts on tau, in both the WT tau:TPR2b (Fig. 5c) and P301L tau:TPR2b (Fig. 5d) XL-MS datasets, the TPR2b cross-link sites recapitulate cross-link patterns observed for TPR2b in the context of full-length DnaJC7. Our data suggest that individual TPR domains bind tightly in a distributed binding mode dictated by TPR2b contacts, suggesting that the arrangement of the three domains within full-length DnaJC7 and their surface properties help define recognition specificity to bind a natively folded conformation of tau.

We next used the MST-binding assay to determine the affinity for each domain to full-length tau. We found that the J domain

has weak binding affinity to WT and P301L tau (Fig. 5e; $8.2 \pm 0.7\,\mu M$ and $28.2 \pm 3.3\,\mu M$, respectively; Supplementary Fig. 5c). In contrast, TPR1 and TPR2b have high binding affinities toward WT tau (Fig. 5e; $84 \pm 6\,nM$ and $75 \pm 5\,nM$, respectively; Supplementary Fig. 5c) that are nearly eight-fold tighter than full-length DnaJC7. Similarly, TPR2a also showed a greater than two-fold tighter binding to WT tau compared with full-length DnaJC7 (Fig. 5e and Supplementary Fig. 5c). We reasoned that the more nonpolar binding groove of TPR2a could impact binding to tau, given its distinct patterning from the other domains (Fig. 5e). For P301L tau binding, we observed a similar increase in binding ability of the individual TPR domains with a greater than twelve-fold increase in binding affinity relative to full-length DnaJC7 (Fig. 5e and Supplementary Fig. 5c).

Given the high-affinity interaction between individual TPR domains and tau, we tested whether individual TPR domains can modulate tau aggregation in vitro via the ThT fluorescence aggregation assay. In these experiments, we incubated $4.4\,\mu M$ full-length tau with 50 nM tau monomer seeds and compared the effects of the addition of equimolar amounts of each TPR domain, or combining all three individual domains, to the full-length DnaJC7 by monitoring tau aggregation for 50 h (Fig. 5f and Supplementary Fig. 5d). The endpoint of the curves showed that tau aggregates upon the addition of tau monomer seeds and that the addition of DnaJC7 completely abolishes tau aggregation. Unexpectedly, none of the individual TPR domains modified tau aggregation significantly and $TPR1_{20-156}$ even yielded a slight increase in aggregation. However, the addition of all three TPRs restores the capacity to inhibit tau aggregation (Fig. 5f and Supplementary Fig. 5d). Thus, while each TPR binds to tau, TPR2b has the highest affinity and the isolated TPR2b domain binds to tau using a recognition groove as detected in the experiments with full-length DnaJC7. Despite this high affinity and maintenance of the binding mode, this single domain alone is insufficient to prevent tau aggregation. These results strongly suggest that DnaJC7 uses TPR2b to recognize tau and that the other domains play a role in the tau anti-aggregation activity.

**Mutations in TPR2b of DnaJC7 prevent tau binding and reverse aggregation-suppression activity**. To directly test our XL-MS-based identification of the tau-binding site on DnaJC7, we designed a DnaJC7 mutant (herein $DnaJC7_{TPR2b\_mut}$) in which we mutated the TPR2b-binding groove. (Fig. 6a). We mutated TPR2b in full-length DnaJC7 at six positions (K260A, N264A, F267A, K268A, E269A, and Y272A) in proximity to cross-link sites (K254, K268, K306, and K329) (Fig. 6b). The $DnaJC7_{TPR2b\_mut}$ mutant bound to full-length WT tau with $6.0 \pm 1.2\,\mu M$ affinity as measured by MST yielding an ~twenty-fold decrease of binding affinity compared with $DnaJC7_{WT}$ that bound with $343 \pm 40\,nM$ (Fig. 6c). We next tested whether $DnaJC7_{TPR2b\_mut}$ retained aggregation-suppression activity in a ThT fluorescence aggregation assay. $DnaJC7_{WT}$ again showed significant suppression of aggregation over 100 h only reaching 20% of the seeded tau-control reaction (Fig. 6d). In contrast, the $DnaJC7_{TPR2b\_mut}$ mutant, which has reduced tau-binding affinity, remained flat for 36 h, after which the signal increased approaching the amplitude of the tau aggregation reactions in the absence of the chaperone (Fig. 6d). The endpoints of the in vitro aggregation reactions were tested in the in-cell tau biosensor assay to quantify the amount of tau aggregates in each sample. The positive (recombinant tau fibrils) and negative (lipofectamine alone) controls yielded $44 \pm 8\%$ and $0.6 \pm 0.1\%$ FRET signal, respectively. The WT full-length tau incubated with tau seed yielded aggregates in $25 \pm 4\%$ of the cells, while addition of $DnaJC7_{WT}$ yielded aggregates in only $1.2 \pm 0.4\%$ of the cells. In

contrast, addition of $DnaJC7_{TPR2b\_mut}$ revealed aggregates in $8.4 \pm 1.6\%$ of cells and approximately seven times more aggregates than $DnaJC7_{WT}$ (Fig. 6e). The TEM images were consistent with the ThT and seeding assay (Supplementary Fig. 6a). Here we validate the XL-MS-identified TPR2b surface on DnaJC7 to be important for binding tau and its efficient tau anti-aggregation activity.

**DnaJC7 recognizes local structures important for the regulation of tau aggregation**. Our high-resolution NMR data, XL-MS, and biochemical experiments on the tau:DnaJC7 complex revealed that DnaJC7 recognizes a small structural element, R1R2, through a binding groove in TPR2b. Recent work from our lab showed that these structural elements, including R1R2, can adopt transient β-hairpin conformations to bury amyloid motifs and that pathogenic mutations perturb these hairpin structures to expose amyloid motifs, thus promoting aggregation[34]. Given that DnaJC7 appears to interact with this structural element, we wondered whether a P270S mutation in the β-turn-stabilizing P–G–G–G motif in the R1R2 element would alter binding affinity to the chaperone. Using MST, we found that P270S tauRD does not bind to DnaJC7 (Supplementary Fig. 7a), indicating that this region is important for DnaJC7 recognition. Further, a ThT fluorescence aggregation assay revealed that DnaJC7 can efficiently control aggregation of WT tauRD but has little-to-no effect on suppressing P270S tauRD aggregation (Supplementary Fig. 7b–d), consistent with our binding measurements. These data support that the R1R2 element is recognized by DnaJC7 in a native-like collapsed conformation.

We used computational modeling to gain more insight into how DnaJC7 can bind the R1R2 tau element. We carried out unbiased coarse-grained docking with DnaJC7 against an ensemble of R1R2 and R2R3 peptides derived from our prior CS-Rosetta tauRD calculations (Fig. 2g). Our ensemble sampled a diversity of R1R2 and R2R3 fragment conformations to produce 5000 models of DnaJC7 bound to each sequence element followed by full-atom refinement to calculate binding energies (Fig. 7a and Supplementary Fig. 7e). We found that many of the predicted binding modes do not yield favorable energetics for binding. However, partitioning the structures according to geometric compatibility with our cross-links revealed R1R2:DnaJC7 complexes with energetically favorable binding modes (Fig. 7b). In fact, simply restricting the models to only ones that are consistent with cross-links between K280 in R1R2 and K254 on TPR2b revealed a set of structures with favorable binding energies. Mapping our cross-link data onto one of these models reveals conformations consistent with our experiments, including the $DnaJC7_{TPR2\_mut}$ construct. The R1R2 element can be stabilized in a collapsed conformation, trapping contacts between K267 and K280/K281 (Fig. 7c, orange dashed lines). Further, K280/K281 forms contacts with K254/K306 of TPR2b (Fig. 7c, yellow dashed lines). The R1R2 element binds to DnaJC7 in a sequence-conserved binding groove (Fig. 7d) in a binding mode that is also compatible with the CSP and peak broadening (Fig. 7e and Supplementary Fig. 2e), suggesting congruency across several experiments to explain how the TPR2b in DnaJC7 can recognize a natively folded conformation of the R1R2 peptide.

Our data predict that the WT R1R2 peptide alone could bind to DnaJC7 and that a P270S R1R2 mutant, which is more unfolded, would not. Though initial efforts to measure the affinity between the peptides and DnaJC7 were unsuccessful, we turned to aggregation experiments to ask if the addition of WT or P270S R1R2 peptides could alter the ability of DnaJC7 to modulate aggregation of tauRD in a ThT fluorescence aggregation assay (Fig. 7f). In the presence of P270S R1R2, we observed no

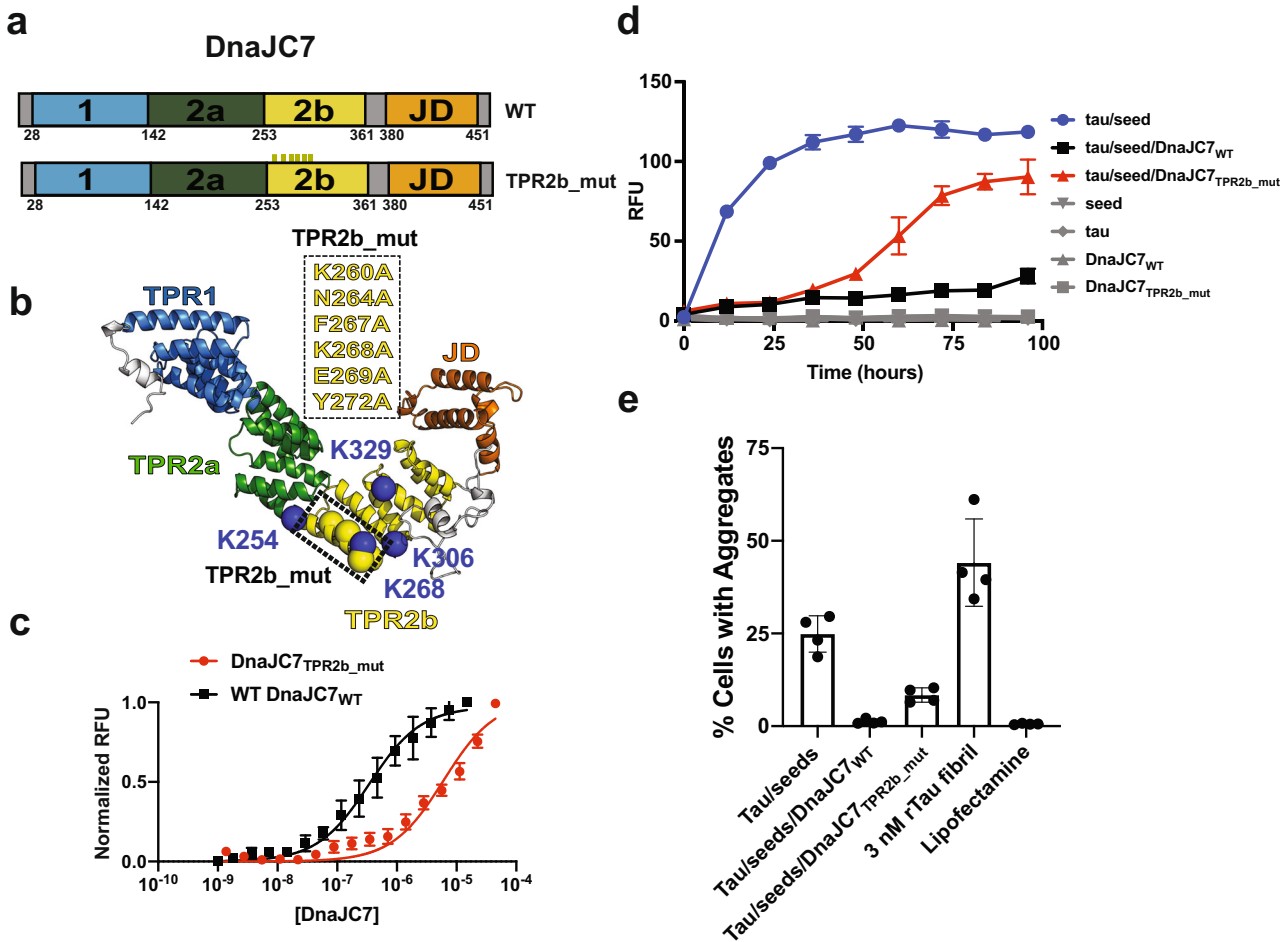

**Fig. 6 Mutations in TPR2b of DnaJC7 disrupt tau binding and aggregation-suppression activity. a** Schematic diagram of the DnaJC7$_{TPR2b\_mut}$ mutant highlighting the location of the six mutations in TPR2b (K260A, N264A, F267A, K268A, E269A, and Y272A). **b** Cartoon representation of DnaJC7$_{TPR2b\_mut}$ oriented to highlight the site of the mutation. Sites of the mutation in TPR2b are shown as yellow spheres. The cross-link sites in TPR2b identified between full-length tau and DnaJC7 are shown as blue spheres (K254, K268, K306 and K329). **c** MST-binding assay for WT tau:DnaJC7$_{WT}$ (black) and WT tau:DnaJC7$_{TPR2b\_mut}$ (red). The MST-binding experiments were performed as technical replicates ($n = 3$) and each concentration is shown as a mean with standard deviation. The data were fit to a linear regression model to estimate the binding constant. **d** 4.4 µM WT tau aggregation was induced using 50 nM tau monomer seeds (blue) in the presence of equimolar DnaJC7$_{WT}$ (black) or DnaJC7$_{TPR2b\_mut}$ (red) chaperone. Gray curves show negative controls: tau alone, DnaJC7$_{WT}$ or DnaJC7$_{TPR2b\_mut}$ alone, and tau monomer seeds alone. Aggregation was monitored using ThT fluorescence. Experiments were performed as technical replicates ($n = 3$) and are shown as mean with standard deviation. **e** The endpoint and control samples of the in vitro aggregation experiment (**d**) were assayed for seeding activity in tau biosensor cells. The in-cell aggregation assay shows significant reduction in seeding in the presence of DnaJC7$_{WT}$ but not in the presence of DnaJC7$_{TPR2b\_mut}$. Tau seeding experiments were performed as biological replicates ($n = 3$) and are shown as mean with standard deviation.

inhibition of DnaJC7 activity (Fig. 7g and Supplementary Fig. 7f). However, the addition of the WT R1R2 peptide allows only 65 ± 5% recovery of DnaJC7 tau aggregation-suppression activity (Fig. 7g and Supplementary Fig. 7f). These data suggest that the WT R1R2 peptide can bind to TPR2b to prevent tau binding to DnaJC7, leading to an increase in tau aggregation, but the P270S R1R2 peptide cannot. These data suggest that DnaJC7 influences tau aggregation by binding to specific elements in tau in a conformationally dependent manner.

## Discussion

How tau changes shape to promote the formation of pathogenic species underlying disease remains an important biological question[42,43]. In the time since the discovery that tau deposits as fibrillar structures in human disease, we have gained insight into the genetics of the disease and uncovered mutations in tau that cause early-onset dominantly inherited tauopathies. Structural insight into how tau mutations change its shape is an important

proxy for understanding how tau can adopt pathogenic conformations. We previously used a multidisciplinary approach to study the structure of a wild-type tau monomeric pathogenic seed and revealed local conformations within tau that underlie amyloid motifs[26]. More recently, we compared the unfolding profiles of WT and P301L tauRD using XL-MS and revealed that pathogenic mutations promote unfolding of the repeat domain in proximity to the P301L mutation, preferentially exposing the $^{306}$VQIVYK$^{311}$ amyloid motif[26,34]. In this study, we compared full-length WT and P301L tau using two parallel cross-linking chemistries to highlight how pathogenic mutations unfold the repeat domain in proximity to the P301L mutation, which impacts the folding of the N-terminus. Our data suggest that tau undergoes discrete conformational changes that underlie disease pathogenesis. While mutations in tau are predominantly linked to frontotemporal dementia, we propose that similar conformational changes that expose amyloidogenic motifs must underlie WT tau conversion into pathogenic seeds. However, the means through

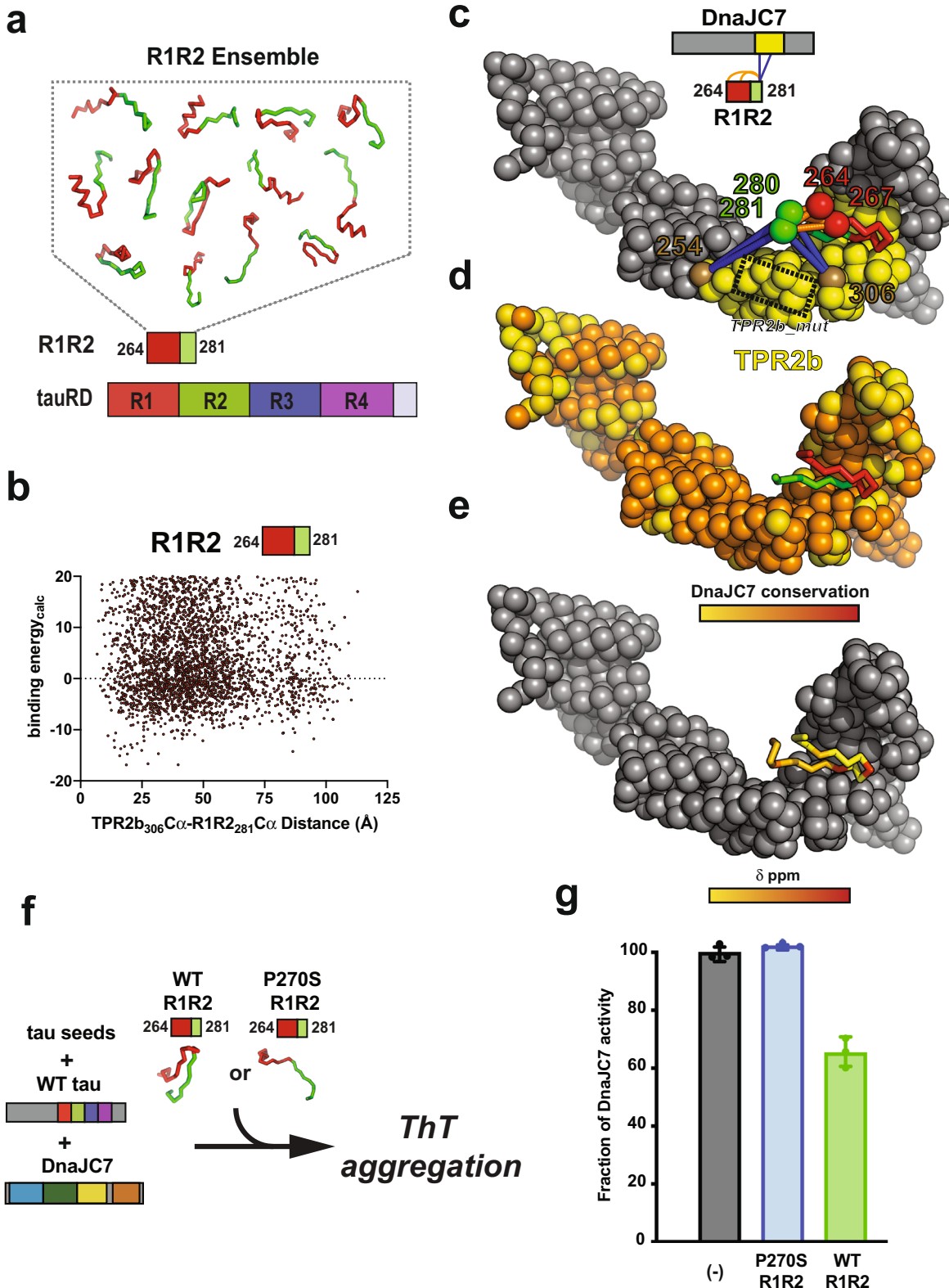

which this happens remains unknown, though cofactors have been proposed as possible drivers of WT tau aggregation. Amyloid motifs central to aggregation are engaged in stabilizing contacts and lead to distinct topologies for fibrils from different diseases[44–47]. Our model predicts that local engagement of the amyloid motifs inhibits aggregation. The early conformational changes enabled by cofactor binding or mutations uncover the

amyloid motifs allowing them to form unique contacts that are on-path to fibril formation.

Our data support that DnaJC7 preferentially binds natively folded forms of tau over more unfolded aggregation-prone mutant forms and tau monomeric seeds derived from WT tau (Fig. 8). To our surprise, DnaJC7 binds to WT tau with 0.4 μM binding affinity, which is tighter than most chaperone:substrate

**Fig. 7 DnaJC7 binds to a natively folded conformation of R1R2. a** Schematic illustrating the generation of an ensemble of R1R2 peptide conformations used in the DnaJC7 docking simulation. The R1R2 peptides are shown in cartoon representation and colored red/green according to the repeat domain. **b** Calculated binding energy and cross-link geometry for the R1R2:DnaJC7 structural ensemble. Each point represents a structural model. Models with low binding energies and short Cα–Cα distances between K306 (DnaJC7) and K281 (tau) were used in subsequent analyses. **c** Representative low-energy scoring model of DnaJC7 bound to the R1R2 peptide. Experimental intramolecular cross-links observed within R1R2 linking K267–K281, K267–K280, K264–K281, and K264–K280, as well as intermolecular contacts between K254(TPR2b):K280/K281(tau) and K306(TPR2b):K280/K281(tau) show congruency in our docking model and XL-MS identified contacts. Location of the DnaJC7$_{TPR2b\_mut}$ is shown in a dashed box. DnaJC7 is shown in spacefill representation and is colored in gray with TPR2b colored in yellow. R1R2 peptide is shown in cartoon representation and is colored in red/green. Cross-link positions are shown in spacefill representation. Intramolecular and intermolecular cross-links are shown as dashed lines and colored in blue and orange, respectively. **d** Mapping the DnaJC7 sequence conservation onto the structural model shows that R1R2 binds in a conserved binding groove in TPR2b. Conservation is colored from yellow to red. **e**. Mapping the CSPs from Fig. 2a onto the R1R2 peptide reveals that the surface on the R1R2 peptide that interacts and is measured by NMR interacts with the TPR2b surface. CSPs are colored from yellow (0 ppm) to red (0.04 ppm). **f, g** Schematic for a competition experiment to determine whether WT R1R2 (collapsed) or a conformational mutant P270S R1R2 (expanded) can bind to DnaJC7 and compete binding to tau, thereby reducing the capacity of DnaJC7 to inhibit tau aggregation in a ThT-aggregation assay. The barplot of the three reactions reveals that the addition of the WT R1R2 peptide (green) can reduce anti-aggregation effect of the DnaJC7 chaperone, while addition of P270S R1R2 (blue) yields a nearly identical signal as no peptide added (black). Experiments were performed as technical replicates ($n = 3$) and are shown as a mean with standard deviation.

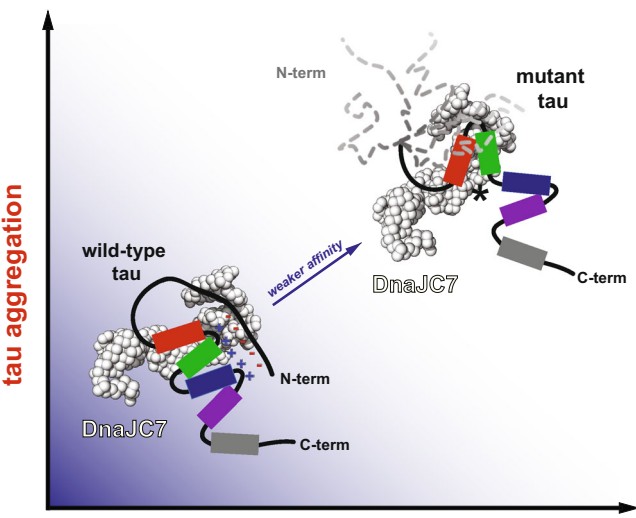

**Fig. 8 Proposed model of DnaJC7 recognition of natively folded conformations of tau to efficiently suppress amyloid aggregation.** Pathogenic mutations in tau unfold local conformations in the repeat domain and influence the conformation of the acidic N-terminal contacts with the more basic repeat domain (shown as "minus" and "plus" symbols), resulting in structural rearrangements that yield an aggregation-prone conformation. DnaJC7 binds to the folded R1R2 element in WT tau and efficiently reduces aggregation. This effect is reduced in the aggregation-prone conformations, such as those found in P301L tau. Tau is colored according to the repeat domains in red, green, blue, and magenta. DnaJC7 is shown in white spheres.

interactions[29,48,49]. However, the general role of chaperones is to recognize nonpolar sequences in proteins that are typically buried in protein cores to promote their folding into functional, folded conformations[1]. Given this high affinity, what is the role of DnaJC7 in tau binding? While many chaperones, including Hsp70s and other JDPs, bind to tau via its encoded amyloid motifs to prevent aggregation, the capacity of these chaperones to suppress tau aggregation is orders of magnitude less efficient[29,50,51]. Furthermore, recent high-resolution NMR studies on substrate interaction with a canonical JDP highlight the chaperone's ability to dynamically recognize nonpolar elements in unfolded substrates through multivalent interactions, likely suggesting a possible mode for prevention of aggregation[52]. In contrast, DnaJC7 encodes an entirely different domain

architecture utilizing TPR domains to recognize natively folded amyloid motif-containing elements in tau. Coupled with its high affinity for tau, this aspect likely underlies DnaJC7's efficient capacity to limit tau aggregation even on long timescales. Finally, our comparison of DnaJC7 activity against WT and P301L tau reveals differences in affinity, likely dictated by changes in tau conformation induced by the pathogenic P301L mutation. Indeed, many pathogenic mutations linked to tau localize to our identified DnaJC7-binding regions, suggesting that DnaJC7 has a reduced capacity to prevent aggregation of more aggregation-prone and disease-associated variants of tau. This perhaps implicates DnaJC7 as an evolved binding partner for tau that may play a role in transporting native conformations of tau by stabilizing aggregation-resistant conformations similarly to how DnaJC7 sequesters the CAR transcription factor in the cytosol. Future cellular experiments are required to reveal the central role of DnaJC7 in tau biology.

Work from Reagan et al. on TPR domains has defined simple rules that set the basis for how these protein-binding modules interact with peptides. However, there exists a large diversity of TPR-containing proteins in mammalian cells and while these rules apply to some, it is not clear how TPR-sequence divergence broadens their capacity to bind structured peptides. In the context of co-chaperones, TPR domains have previously been proposed to recognize conserved EEVD-sequence motifs on C-terminal tails on Hsp70 and Hsp90 chaperones[53]. For example, HOP uses two of its three TPR modules to simultaneously bind to the acidic tails of these proteins and thus bring them together[54]. Details for this mechanism are not clear, but it is thought that HOP may promote the transfer of the substrate from one chaperone to the other. However, DnaJC7 has not been implicated in binding to protein clients directly[55]. While the three TPR-domain architecture of DnaJC7 is similar to HOP, DnaJC7 also encodes a C-terminal J domain. Mechanistic insight on DnaJC7 is sparse in the literature, though peptide-binding experiments have shown that DnaJC7 can bind Hsp70/Hsp90-derived EEVD sequences with weak affinity[56]. Alignment of the HOP and DnaJC7 sequences reveals little similarity, including sequence changes in residues on TPRs important for recognition of EEVD-containing peptides. High sequence conservation of the DnaJC7 TPR domains in metazoans suggests that this chaperone may have additional functions divergent from HOP. It is possible that DnaJC7 can bind directly to substrates through the TPR domains (tau via TPR2b) and that it can also recruit Hsp70 and Hsp90 through their acidic tails to localize substrates to these chaperones. Hsp90 and Hsp70 are both known to be associated with

tau[51,57,58]. Thus, in the context of DnaJC7, it is possible that direct substrate binding via TPR domains allows the C-terminal J domain to recruit Hsp70 to promote substrate transfer. Resolving the potential dual nature of substrate binding and chaperone recruitment will require additional experiments, but this complex model highlights a more nuanced cooperation of chaperones in networks to maintain a healthy proteome.

## Methods

**Isolation of tau from PS19 mouse brains**. Soluble tau species were immuno-purified from PS19 mouse brain using the HJ8.5 anti-human tau antibody[26]. Each time point (weeks 1 through 6) involved three mouse brains ($n = 3$). Briefly, 0.5 g of brain tissues from week-1, -2, -3, -4, -5, and week-6 mouse were gently Dounce-homogenized in 1X PBS buffer in the presence of protease inhibitors and centrifuged at 21,000 RCF for 15 min at 4 °C to remove the debris. The supernatants were mixed with HJ8.5 (1:50, m/m) and 50% slurry protein G-agarose beads (1:5, v/v) and incubated at 4 °C overnight. Next day, the binding reactions were centrifuged at 1000 RCF for 3 min and the pellet was washed with Ag/Ab-binding buffer (Thermo Scientific) three times. The antibody-bound tau was eluted in 100 μL of low-pH elution buffer directly into 10 μL of Tris-base, pH 8.5, to neutralize the buffer. Elution was repeated once more with 50 μL of elution buffer into 5 μL of Tris-base buffer, pH 8.5, for a total volume of 165 μL. Samples were further purified on a Superdex 200 Increase 10/300 GL column on an AKTA FPLC. Fractions containing tau were determined using western blot and the pooled samples were quantified with a Micro BCA assay (Thermo Scientific), flash-frozen in liquid nitrogen, and stored at −80 °C.

**Mass spectrometry analysis of mouse tau samples**. The purified tau samples were denatured with 8 M urea, reduced with 2.5 mM TCEP, alkylated with 5 mM iodoacetamide in the dark for 30 min, and the urea diluted to 1 M using 50 mM ammonium bicarbonate. Trypsin was added to 1:50 (m/m) (Promega) and incubated overnight at 37 °C with 600 rpm shaking on a TherMomixer® C (Eppendorf). In all, 2% (v/v) formic acid was added to acidify the reaction and further purified by reverse-phase Sep-Pak tC18 cartridges (Waters). The eluted peptides were quantified on a DeNovix nanospectrometer. The dried samples were resuspended in water/acetonitrile/formic acid (95:5:0.1, v/v/v) to a final concentration of approximately 0.5 μg/μL. In all, 2 μL each were injected into an Eksigent 1D-NanoLC-Ultra HPLC system coupled to a Thermo Orbitrap Fusion Tribrid system at the UTSW Proteomics core. The data for each fraction were analyzed using the Proteome Discoverer Suite v2.4 (Thermo) and searched against the mouse proteome to identify hits. Only hits with PSM values of >5 were further considered. The cumulative intensities for each hit were normalized to MAPT intensities. Final analysis focused on all 47 members of the JDP family. JDP abundance was normalized to intensity values for 1N4R tau according to the following equation: [$I_{JDP}/I_{tau}$]*100.

**Co-immunoprecipitation of tau with DnaJC7**. Littermate wild-type and PS19 mouse brains weighing approximately 400 mg were Dounce homogenized in 5 mL of TBS (10 mM Tris, 150 mM NaCl, pH 7.4). Homogenates were then clarified by centrifugation at 4 °C at a speed of 17,200 RCF for 15 min. The supernatant was then isolated. Total brain protein concentrations were measured using the Pierce™ BCA Protein Assay Kit (ThermoFisher). For immunoprecipitations, the magnetic Dynabeads™ Protein A Immunoprecipitation Kit (ThermoFisher) was used following the standard protocol from ThermoFisher. 4 mg of anti-DnaJC7 antibody (Proteintech, 11090-1-AP) was conjugated to 50 μL of total Dynabeads™ suspension overnight at 4 °C with rotation. Normal rabbit IgG (4 mg, Abcam, ab37415) was used for control immunoprecipitations. 300 μg of total mouse brain protein was added to the antibody-conjugated Dynabeads™ and incubated at 4 °C overnight. After overnight incubation, the flow-through was collected after beads were isolated using a DynaMag™ Magnet. Following washes of the beads with Washing Buffer, proteins were incubated with 20 μL of Elution Buffer for 5 min at room temperature. Proteins were then eluted by collecting the supernatant after beads were isolated using a DynaMag™ Magnet.

Immunoprecipitation elutions and flow-throughs, mouse brain homogenates, and recombinantly purified 2N4R tau and DnaJC7 samples were prepared in 1X (final) LDS Bolt™ buffer (Invitrogen) supplemented with 10% β-mercaptoethanol and heated for 10 min at 98 °C. The samples consisted of 10 μL of IP elutions, 1 or 10% (Tau blot and DnaJC7 blot, respectively) of the total brain protein input into the IP, 1 or 10% (Tau blot and DnaJC7 blot, respectively) of the collected flow-through volume, and 50 ng of recombinant Tau or DnaJC7. The proteins were resolved by SDS-PAGE using Novex NuPAGE precast gradient Bis-Tris acrylamide gels (4–12%) (Invitrogen). After gel electrophoresis, resolved proteins were transferred onto Immobilon-P PVDF membranes (Millipore Sigma) using a Bio-Rad Trans-blot® semidry transfer cell. After protein transfer, membranes were blocked in TBST buffer (10 mM Tris, 150 mM NaCl, pH 7.4, and 0.05% Tween-20) containing 5% nonfat milk powder (Bio-Rad). Membranes were then probed with antibody in TBST containing 5% milk powder. The following antibodies were used for immunoblotting: rabbit polyclonal anti-DnaJC7 (Proteintech, 11090-1-AP) at a

1:2000 dilution, rabbit polyclonal anti-tau (Agilent, A002401-2) at a 1:3000 dilution, a Veriblot secondary antibody that only recognizes native (nondenatured) antibodies (Abcam, ab131366) at a 1:2000 dilution to minimize the detection of the heavy and light chains from the antibody used in the immunoprecipitation when blotting for DnaJC7, and a secondary donkey-anti-rabbit HRP-linked F(ab')$_2$ at a 1:8000 dilution (Cytiva, NA9340-1ML) when blotting for tau.

**CRISPR/Cas9 knockout of DnaJC7 in tau biosensor cells**. Four human gRNA sequences per gene were selected from the Brunello library[59]. A single nontargeting human gRNA sequence was used as a negative control. For all gRNA sequences not beginning with guanine, a single guanine nucleotide was added at the 5′-end of the sequence to enhance U6 promoter activity. DNA oligonucleotides were synthesized by IDT DNA and cloned into the lentiCRISPRv2 vector[60] for lentivirus production. The plasmids for the four gRNAs for each gene were pooled together and used to generate lentivirus.

Lentivirus was produced as described previously[61]. HEK293T cells were plated at a concentration of 100,000 cells/well in a 24-well plate. About 24 h later, cells were transiently cotransfected with PSP helper plasmid (300 ng), VSV-G (100 ng), and gRNA plasmids (100 ng) using 1.875 μL of TransIT-293 (Mirus) transfection reagent. About 48 h later, the conditioned medium was harvested and centrifuged at 1,200 RCF for five minutes to remove dead cells and debris. For transduction, 30 μL of the virus suspension was added to HEK293T tau biosensor cells at a cell confluency of 60% in a 96-well plate. About 48 h post transduction, infected cells were treated with 1 μg/ml puromycin (Life Technologies, Inc.) and maintained under puromycin selection for at least ten days after the first lentiviral transduction before conducting experiments.

**Flow cytometry of tau biosensor cells**. Samples from immunoprecipitated mouse brain or ThT assay at $T = 0$ (directly from the freezer) and $T = 6$ days (endpoint of the aggregation experiment) were assayed for their seeding activity in HEK293T tau biosensor cells. For the DnaJC7 KO and nontargeting control seeding experiments, HEK293T tau biosensor cells with either a DnaJC7 KO or a nontargeting control were used. For all experiments, cells were plated in 96-well plates at 20,000 cells per well in 100 μL of media. About 24 h later, the cells were treated with 50 μL of a heparin-induced recombinant tau-fibril dilution series. Prior to cell treatment, the recombinant tau fibrils were sonicated for 30 seconds at an amplitude of 65 on a Q700 Sonicator (QSonica). A three-fold dilution series of the sonicated fibril concentrations ranging from 100 nM to 15.2 pM and a media control was added to the cells. About 48 h after treatment with tau, the cells were harvested by 0.05% trypsin digestion and then fixed in PBS with 2% paraformaldehyde.

A BD LSRFortessa SORP was used to perform FRET flow cytometry. To measure mCerulean and FRET signal, cells were excited with the 405-nm laser and fluorescence was captured with a 405/50-nm and 525/50-nm filter, respectively. To measure mClover signal, cells were excited with a 488-nm laser and fluorescence was captured with a 525/50-nm filter. To quantify FRET, we used a gating strategy where mCerulean bleed-through into the mClover and FRET channels was compensated using FlowJo analysis software, as described previously[34]. The gating strategy for the tau biosensor flow cytometry analysis to quantify the percentage of cells that have tau aggregates and are FRET positive is highlighted in Supplementary Fig. 1e. FRET signal is defined as the percentage of FRET-positive cells in all analyses. For each experiment, 10,000 cells (SSC singlets, see Supp. Fig. 1e) per replicate were analyzed and each condition was analyzed as biological triplicates. Data analysis was performed using FlowJo v10 software (Treestar).

**Protein expression and purification**. Wild-type and mutant full-length tau or tauRD were purified from *E. Coli* BL21 (DE3) transformed with pET28b plasmid using the same protocol as described previously[34]. N[15]-labeled human tauRD was expressed in M9 medium supplemented with 2 mM MgSO$_4$, 0.1 mM CaCl$_2$, 0.4% (w/v) D-glucose, 0.0005% (w/v) thiamine, and trace elements, and was purified with the same procedure as unlabeled protein. The production of tau seeds was carried out by incubating 16 μM wild-type tau with 1:1 molar ratio of Heparin (AMSbio) for 1 h at 37 °C in 30 mM MOPS pH 7.4, 50 mM KCl, 5 mM MgCl$_2$, and 1 mM with DTT (MOPS buffer). The tau:heparin reactions were injected onto a Superdex 200 Increase 10/300 GL Column (GE) in 1X PBS yielding a peak that eluted around 1 ml earlier than wild-type tau. The seeding activity was confirmed using tau FRET biosensor cells.

The pMCSG7-DnaJC7 plasmid was a kind gift from Dr. Andrzej Joachimiak (Argonne National Lab). Briefly, harvested cells were lysed by a pressure homogenizer in a buffer containing 50 mM Tris, pH 7.4, 500 mM NaCl, 1 mM β-mercaptoethanol, 20 mM imidazole, and 1 mM phenylmethylsulfonyl fluoride (PMSF). The clarified cell lysate was incubated with Ni-NTA beads at 4 °C for one hour, and the bound protein was eluted with liner gradient of 20–500 mM imidazole. The pooled fractions were buffer-exchanged into 50 mM Tris, pH6.0, 20 mM NaCl, and 2 mM DTT. The sample was further purified using a HiTrap Q column (GE Healthcare Life Sciences) followed by HiLoad 16/600 Superdex 200 pg gel-filtration chromatography (GE Healthcare Life Sciences) and eluted in MOPS buffer. The protein was then concentrated, concentration quantified on a nanospectrometer, aliquoted, flash-frozen in liquid nitrogen, and stored at −80 °C. The DnaJC7$_{TPR2b\_mut}$ construct was cloned into pET29b into the NdeI and XhoI

sites (Twist Biosciences) and was expressed and purified using the same protocol as WT DnaJC7.

Single TPR domains and the J domain were cloned using Gibson assembly into pMCSG7 plasmid. Protein expression was induced the same as full-length DnaJC7 (see above). The TPR domains were purified under denaturing conditions and refolded on Ni-NTA beads. Cell pellets were treated with 8 M urea and the lysate was centrifuged to remove debris. Denatured and clarified lysates were incubated with Ni-NTA beads at 4 °C for one hour. The Ni-NTA beads were washed with a decreasing gradient of urea using a 50 mM Tris, pH 7.4, 500 mM NaCl, and 1 mM β-mercaptoethanol base buffer. Finally, the beads were washed with 50 mM Tris, pH 7.4, 500 mM NaCl, and 1 mM β-mercaptoethanol. Samples were eluted with 300 mM imidazole in 50 mM Tris, pH 7.4, 500 mM NaCl, and 1 mM β-mercaptoethanol and applied to a Superdex 75 Increase 10/300 column (GE) after concentration. Then samples were eluted in 30 mM MOPS, pH 7.4, 50 mM KCl, 5 mM MgCl₂, and 1 mM with DTT. The purified individual TPR domains were confirmed to be monomeric using SEC, dynamic light scattering, and cross-linking coupled to SDS-PAGE. The protein was then concentrated, concentration quantified on a nanospectrometer, aliquoted, flash-frozen in liquid nitrogen, and stored at −80 °C.

**Microscale thermophoresis**. MST experiments were performed on Nanotemper Monolith NT.115 in the Molecular Biophysics Resource core at UTSW and analyzed with a standard protocol[62]. All binding measurements were done as technical triplicates. Wild-type or mutant tau/tauRD was labeled with Cyanine5 NHS ester dye (Cy5) and titrated by a serial two-fold dilution of DnaJC7 or single DnaJC7 domains. For peptide binding, DnaJC7 was instead labeled with Cy5. Data were fit in PALMIST in a 1:1 binding model and analyzed in GUSSI[62]. The binding stoichiometry for WT tau:DnaJC7 and P301L tau:DnaJC7 was assessed by comparing the binding curves using data from the T-jump or across the entire range[63] and verified using cross-linking of complexes and visualized by SDS-PAGE.

**Cross-linking mass spectrometry analysis**. We have developed standardized protocols for cross-linking and data analysis of samples. For DSS reactions, full-length tau or tauRD were cross-linked at 1 mg/ml in 100 µL total volume with a final 1 mM DSS (DSS-d₀ and -d₁₂, Creative Molecules) for one minute at 37 °C while shaking at 350 rpm on a ThermoMixer® C (Eppendorf). For ADH/DMTMM reactions, protein samples were incubated with 57 mM ADH (d₀/d₈, Creative Molecules) and 36 mM DMTMM (Sigma-Aldrich) for 15 min at 37 °C while shaking at 350 rpm on a ThermoMixer® C. For complexes between full-length tau/tauRD and DnaJC7/TPR domains, we incubated tau with the chaperone at a 1:1.2 molar ratio for 1 h at 25 °C followed by incubation with 1 mM DSS (DSS-d₀ and -d₁₂, Creative Molecules) for one minute at 37 °C. The reactions were quenched with 100 mM ammonium bicarbonate (AB) for 30 min. Samples were resolved on SDS-PAGE gels (NUPAGE™, 4–12%, Bis-tris, 1.5 mm, or homemade SDS-PAGE gel) and bands corresponding to tauRD monomer, tau monomer, tauRD:DnaJC7 heterodimer, tau:DnaJC7 heterodimer or tau:TPR domain heterodimers were gel-extracted following standard protocols[34]. Samples were flash-frozen in liquid nitrogen, lyophilized, and resuspended in 8 M urea followed by 2.5 mM TCEP reduction and 5 mM iodoacetamide alkylation in the dark with each 30 min. Samples were then diluted to 1 M urea by 50 mM AB and digested by 1:50 (m/m) trypsin (Promega) by overnight shaking at 600 rpm on a ThermoMixer® C. About 2% (v/v) formic acid was added to acidify the reaction system and further purified by reverse-phase Sep-Pak tC18 cartridges (Waters) and size-exclusion peptide chromatography (SEPC). The fraction collected from SEPC was lyophilized. The dried samples were resuspended in water/acetonitrile/formic acid (95:5:0.1, v/v/v) to a final concentration of approximately 0.5 µg/µL. About 2 µL of each was injected into Eksigent 1D-NanoLC-Ultra HPLC system coupled to a Thermo Orbitrap Fusion Tribrid system at the UTSW Proteomics core.

The analysis of the mass spectrum data was done by an in-house version of xQuest[37]. Each Thermo.raw data file was first converted to open.mzXML format using msconvert (proteowizard.sourceforge.net). The mass spectra across replicates yielded similar intensities (Source Data 2). Search parameters were set differently based on the cross-link reagent as follows: for DSS, the maximum number of missed cleavages (excluding the cross-linking site) = 2, peptide length = 5–50 aa, fixed modifications = carbamidomethyl-Cys (mass shift = 57.021460 Da), mass shift of the light cross-linker = 138.068080 Da, mass shift of mono-links = 156.078644 and 155.096428 Da, and MS1 tolerance = 10 ppm, MS2 tolerance = 0.2 Da for common ions and 0.3 Da for cross-link ions, search in ion-tag mode. For zero-length cross-link search: maximum number of missed cleavages = 2, peptide length = 5–50 residues, fixed modification carbamidomethyl-Cys (mass shift = 57.02146 Da), mass shift of cross-linker = −18.010595 Da, no monolink mass specified, MS1 tolerance = 15 ppm, and MS2 tolerance = 0.2 Da for common ions and 0.3 Da for cross-link ions; search in enumeration mode. For DMTMM zero-length cross-link search: maximum number of missed cleavages = 2, peptide length = 5–50 residues, fixed modifications = carbamidomethyl-Cys (mass shift = 57.02146 Da), mass shift of cross-linker = −18.010595 Da, no monolink mass specified, MS1 tolerance = 15 ppm, and MS2 tolerance = 0.2 Da for common ions and 0.3 Da for cross-link ions; search in enumeration mode. For ADH: maximum number of missed cleavages (excluding the cross-linking site) = 2, peptide length = 5–50 residues, fixed

modifications = carbamidomethyl-Cys (mass shift = 57.021460 Da), mass shift of the light cross-linker = 138.09055 Da, mass shift of monolinks = 156.10111 Da, MS1 tolerance = 15 ppm, and MS2 tolerance = 0.2 Da for common ions and 0.3 Da for cross-link ions, search in ion-tag mode. False discovery ratios were estimated by xprophet[64] to be 0–0.17%. For each experiment, five replicate datasets were compared and only cross-link pairs that appeared in at least three datasets were used to generate a consensus dataset (Source Data 2). We have developed the use of a consensus dataset across technical replicate datasets to help interpret behavior of cross-links in noisy XL-MS datasets ideally suited for IDPs and other highly dynamic systems. For all analyses, the consensus datasets were used to infer changes in cross-link frequency or distribution. The pair position and unique seen numbers (frequency) were visualized using a custom gnuplot script.

**¹⁵N-¹H TROSY-HSQC**. Two-dimensional ¹⁵N-¹H TROSY-HSQC spectra were recorded on Agilent DD2 600-MHZ spectrometers at the UT Southwestern Biomolecular NMR Facility. The ¹⁵N-labeled tauRD and DnaJC7 were both buffer-exchanged into 10 mM Na₂HPO₄, pH 7.4, 100 mM NaCl, and 4 mM DTT with 8% D₂O. Each HSQC run was performed at 10 °C with either 100 µM ¹⁵N-labeled tauRD alone or titrated by DnaJC7 with molar ratios at 1:0.5, 1:1, and 1:2. The spectrum was converted and phase-corrected using NMRPipe[65]. Peak assignments were based on the deposited information from BMRB (19253) and unpublished data from Dr. Guy Lippens. The software Sparky was used to analyze chemical shift perturbations and peak-intensity changes across concentrations[66].

**Thioflavin-aggregation assay**. Wild-type full-length tau or tauRD (or mutants) were diluted to 17.6 µM in MOPS buffer with 25 µM β-mercaptoethanol and boiled at 100 °C for 5 min. The proteins were further diluted two-fold in PBS and ThT was added to a final concentration of 25 µM in the dark. For a 60 µL reaction system, 30 µL of tau or tauRD protein was mixed with an equal volume of a mixture consisting of either buffer, seeding monomer, DnaJC7, tauRD, tau, WT R1R2, or P270S R1R2 or any combination of these[34]. All experiments were performed as biological triplicates. ThT kinetic scans were run every 10 min on a Tecan Spark plate reader at 446-nm Ex (5-nm bandwidth), 482-nm Em (5-nm bandwidth) with agitation for 5 s prior to acquisition.

**Transmission electron microscopy**. 5 µL of sample was loaded onto a glow-discharged Formvar-coated 200-mesh copper grid for 30 s and was blotted by filter paper followed by washing the grid with 5 µL of ddH₂O. After another 30 seconds, 2% uranyl acetate was loaded on the grid and blotted again. The grid was dried for 1 min and loaded into a FEI Tecnai G2 Spirit Biotwin TEM. All images were captured using a Gatan 2K × 2K multiport readout post-column CCD at the UT Southwestern EM Core Facility.

**Modeling tau and its interaction with DnaJC7**. Modeling of tauRD was performed using CS-ROSETTA guided by backbone chemical shifts kindly provided by Dr. Guy Lippens. In total, 5000 models of tauRD were produced on the BioHPC cluster at UTSW. Chemical shift perturbations were mapped onto a representative model using an in-house python script. R1R2 or R2R3 sequence elements were extracted from the 5000-model ensemble and the conformations were structurally aligned to produce a small set of unique conformation using the cluster protocol in Rosetta. Models of DnaJC7 were built by homology modeling by employing a DnaJC3 structural model as a template. The resulting models were relaxed and the lowest-scoring models were selected for subsequent analysis[38]. The EnsembleDock protocol[67] allowed the docking of an ensemble of R1R2 and R2R3 peptide conformations (from above) against an ensemble of DnaJC7 conformations using first a low-resolution centroid mode, followed by full-atom relax, resulting in an unrestrained ensemble of DnaJC7 bound to R1R2 or R2R3 peptides. The change in binding energy was calculated using the ddG protocol to evaluate the predicted binding energy[26]. Additionally, for each model in the ensemble, we computed the distance between Cα–Cα atoms between tau:DnaJC7 for positions that cross-linked in the XL-MS experiments using an in-house python script[34]. The predicted binding energy was plotted as a function of the cross-link distance geometry to identify models with low energy that satisfied our experimental data. All simulations were performed on UTSW's BioHPC computing cluster. All plots were generated with gnuplot. Images were created using PyMOL.

**Reporting summary**. Further information on research design is available in the Nature Research Reporting Summary linked to this article.

## Data availability

Raw mass spectrometry data for tau immunoprecipitation are available in Source Data 1. Raw cross-linking mass spectrometry data are available in Source Data 2. All raw data for western blots, FRET tau biosensor analysis, MST-binding measurements, NMR chemical shift analysis, tauRD Rosetta simulations, ThT fluorescence aggregation experiments, and Rosetta docking simulations are all available in Source Data 3. Other supporting data are available upon reasonable request from the authors. Source data are provided with this paper.

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

## Acknowledgements

This work was supported by grants to L.A.J from the Marie Effie Cain Endowed Scholarship, a Chan Zuckerberg Initiative Collaborative Science Award (2018-191983), and a Bright Focus Foundation grant (A2019060). We appreciate the help of the Molecular Biophysics Resource core, Structural Biology Laboratory, Biomolecular Nuclear Magnetic Resonance Facility, Cryo-Electron Microscopy Facility, and Proteomics Core Facility at the University of Texas Southwestern Medical Center. We thank Dailu Chen for helping in analysis of the XL-MS monolink and looplink data. We also thank members of the Joachimiak lab for reading and providing critical comments on the paper.

## Author contributions

Z.H. and L.A.J. conceived and designed the overall study. V.A.P. performed the immunoprecipitations and seeding experiments in WT and KO biosensor cells. Z.H., P.M.W., and O.K. purified all the recombinant proteins. Z.H. performed in vitro protein binding and aggregation assays. Z.H. and B.D.R. acquired and analyzed the NMR spectra. Z.H. and P.M.W. performed the cross-link mass spectrometry experiments and analysis. Z.H. performed the ROSETTA simulations. A.M.O. performed the in-cell seeding experiments. B.D.R. performed the TEM. Z.H. and L.A.J. wrote the paper, and all authors contributed to its improvement.

## Competing interests

The authors declare no competing interests.
