## [Peer Review File · Nature Communications]

DnaJC7 binds natively folded structural elements in tau to inhibit amyloid formationREVIEWER COMMENTS

Reviewer #1 (Remarks to the Author):

In this work, the authors explore differential binding of proteins to tau during its maturation as an aggregation seed. Mass spectrometry analysis of these samples suggested that a number of chaperones, including Hsp40s, interact with tau. The authors focus on those that change association with tau during this time, leading them to DnaJC7 (although DnaJC5 is also quite interesting). They then proceed to study this interaction in more detail, suggesting that DnaJC7 binds the microtubule-binding repeats R1 and R2 (Fig 2), prefers the WT state (Fig 4) and that the interaction may occur through all three TPR domains (Fig 5). Perhaps most strikingly, they show that CRISPR knockdown of DnaJC7 increases seeding (Fig 1C). Although not a major focus of the work, the observed difference in crosslinks between WT and P301L full length tau proteins (in the absence of DnaJC7) is a nice addition (fig 3c, e), as it extends from the author's previous work on shorter, RD versions. Overall, this is an interesting and important work that suggests a role for DnaJC7 in tau aggregation. The analysis of tau-associated protein during seeding in Figure 1 and the Supplemental is also likely of broad interest. However, some of the interpretations seem premature and there needs to be more rigorous controls in places.

Major:

1. An alternative, simple explanation for the difference in DnaJC7's affinity between full length and RD tau is that there are additional, low affinity binding sites for the chaperone in the N- and C-terminal regions, adding up to a tighter apparent affinity. What is the stoichiometry of the DnaJC7-tau complex in both cases? Hill slope from the MST? Stoichiometry by SEC-MALS? Chaperones are notoriously "sticky" so it would not be surprising to find secondary sites. This is an important question because another simple reason for the change in apparent affinity between WT and P301L is a change in stoichiometry. (i.e. distinct from the current interpretation of a change in tau "structure").
2. In the crosslinking experiments with DnaJC7 (Fig 3), it is not clear if the chaperone is present at its Kd concentration, as the two tau constructs have different affinity values? Alternatively, if the concentration of DnaJC7 is saturating, then why are there differences observed? This experiment needs to be designed to allow thermodynamic comparisons between the two samples and it was difficult to parse the required details from the legends or methods.
3. Related to this comment, is there sufficient time in the crosslinking experiments to allow for saturable DnaJC7 binding before the crosslinking reactions? As above, there needs to be a careful examination of kinetic parameters.
4. More work is needed to ensure that DnaJC7 TPR truncations (TPR1 20-156, etc) are folded. It seems that the results could be explained by differences in the solubility, aggregation or folding of the truncations. Are they folded, as judged by circular dichroism? Are they monomeric on SEC (esp at the concentrations and buffers and times and temperatures employed)? This question arises, in part, from the counter-intuitive and somewhat confusing observation that some of the truncations have a tighter apparent affinity than the full length. There is either an auto-inhibitory mechanism (which would be unexpected amongst the TPR proteins) or, more likely, some of the constructs are (at least partially) aggregated.
5. Related to this comment, the truncation of TPR domains doesn't seem like the best way to address the question of where the tau interaction is taking place. With the modelling (Fig 6), point mutations could be made in the context of full length DnaJC7 to test this idea more specifically.
6. The observation that P270S tauRD does not bind DnaJC7 is interesting, but it is not clear whether it makes thermodynamic sense? What is the energetic difference between the folded and unfolded state? It seems unlikely to be very large, given that the protein is an IDP. So, wouldn't the P270S sample both states – and therefore a portion of the protein be accessible for binding DnaJC7? Tau has few residues for easy protein concentration determination (e.g. Trp), so one wonders if the concentrations of the two samples are just different? Or P270S tau RD is mildly aggregation prone under these conditions and not as available for binding? The interpretation of this part of the manuscript (Fig 6f, g)

seems speculative.

7. Does DnaJC7 block the aggregation of P301L tau seeded with WT aggregates (as in Figure 4)? That experiment would discern whether the difference in activity of DnaJC7 between the WT and P301L was due to chaperone binding to the monomer or the seed. The uniform lag time in the ThT assays (see Fig 4A, B) and the equimolar requirement suggest that the binding activity is on monomer, but the mixed seeding experiment would finalize that idea.

8. Throughout the figures, the individual data points for replicates should be shown to align with Nature/Springer standards. This is especially important for some of the more subtle differences (Fig 2b, Fig 4C, D). Also, it needs to be clear if replicates are independent (e.g different batches of protein) or technical.

Minor:

1. The argument that the MD results align with the crosslinking (page 16, line 400) is circular, as the crosslinking was used to guide/restrict the docking.

2. The sentence on page 4, line 66-68 seems redundant with others in the paragraph.

3. Hsp40s is no longer the preferred nomenclature for the J-domain proteins (Kampinga et al. 2019 Cell Stress Chaperones), largely because members of this class of proteins have molecular mass values that are not uniformly 40 kDa. For example, DnaJC7 is ~55 kDa.

4. The use of "dramatically" and "high affinity" in page 12 (line 291 and 296, respectively) don't seem justified by the 2-fold changes in crosslinking and the high nanomolar affinity constants.

5. How many significant figures are appropriate for the cell-based seeding assay? Currently, the results are reported as three ($69.7 \pm 5.3 \%$), but given the error, it seems that two might be more appropriate (e.g. $70 \pm 5.3 \%$).

Reviewer #2 (Remarks to the Author):

This is an interesting work that reports the interaction between DnaJC7 and tau. The authors extended their previous work on tau and here they used a combined approach to investigate the binding of tau with DnaJC7 and how the chaperone inhibits tau's aggregation. They found out that several chaperones, including an number of DnaJs (Hsp40s) engage tau. They decided to focus on DnaJC7 given that no substrates have been reported for this protein. Cross-linking Mass Spectrometry was employed to probe for the interaction, complemented with NMR, binding assays and finally coarse-grained docking. This is arguably a challenging system and thus a high-resolution structure of the complex cannot be easily obtained. Nevertheless, the insight provided by this work will be of great interest to the community that justifies publication at Nat Comm. It is particularly intriguing that the unfolded variant of tau appears to bind with lower affinity relative to the "more folded" wild type tau. The experiments are very well executed, clearly described and the findings efficiently presented in the text and in the figures. I only have some minor comments that hopefully can provide some clarity and improve the manuscript.

1. Do the authors have any data regarding the binding of the N-terminal region of tau to DnaJC7? Do they know where on DnaJC7 it binds?

2. It is surprising that the J domain seems to bind to tau, albeit with low affinity. Still an 8 microM affinity is not trivial. Is this within experimental noise or the authors believe there is indeed binding?

3. In the NMR titration experiment of labeled tau by DnaJC7, several resonances broaden beyond detection. What are these residues? And what is the reason for experiencing line broadening? Is it because of their direct interaction with DnaJC7? In this case, wouldn't this be suggestive of a stronger binding compared to the other residues that simply display chemical shift perturbation? Is there assignment for these residues?

4. The authors state that there is very little information about how Hsp40s recognize and interact with substrates. I am thus surprised that the authors did not mention or discuss a key structure

determined recently of a full-length Hsp40 in complex with an unfolded substrate (PMID 31604242). It would be of interest if the authors compared the mode of binding reported here with the structure of the Hsp40 reported previously, even if the binding domains are different.

Reviewer #3 (Remarks to the Author):

The manuscript from Hou et al nicely illustrated the role of DnaJC7 in stabilizing natively folded tau to prevent tau conversion. However, I still have a few concerns, especially to the data quality of the quantitative crosslinking analysis. My points are as follows.

1. My biggest concern is related to the crosslink comparison between different species, i.e. WT tau, P301L tau, WT tau+DnaJC7 and P301L tau+DnaJC7. I am puzzled that all analyses were done based on crosslink numbers, rather than standard quantitative MS analysis. Given that the authors acquired many replicates per sample, it would be much more accurate to use label free quantitative pipeline rather than merely counting crosslink numbers. Therefore I would suggest the authors to re-analyze their crosslinking data using label free quantitation.

2. The authors extracted the gel bands corresponding to the species of interest for crosslinking analysis. Did they make sure that the same amount of peptides were loaded on LCMS, or were some kind of normalization method applied between samples? Since only crosslink counts were considered for quantification and the number of crosslinks vary significantly depending on the sample load, I am skeptical about how accurate the quantification is in the whole crosslinking analysis. As shown in Supplementary 3b, the intensities of Coomassie staining are rather different in different samples. Especially for the DNAJC7+tau heterodimer, I could barely see any gel bands at the corresponding molecular weight.

3. In general, I found the crosslink plots shown as supplementary figure c-e are difficult to read. Surely I can see the differences in the boxed areas (for instance the dashed boxes between WT and P301L tau) but I also see changes in other regions of the plot. I think the authors should provide either a cutoff value or some statistical measurements to distinguish significant from non-significant changes. Another point is in supplementary figure e, the authors claim a complete absence of long range contacts with the N-terminus of WT tau (from WT tau+DnaJC7 complex), but I do see a dot connecting residues between 0-50 to 350-400. Am I taking something wrong here?

4. What is the reason that crosslinking was performed at 50 °C? The authors stated in the paper “we employed these two temperatures to probe the stability of the intensities in response to unfolding”, but I didn’t find any data supporting how temperature affects unfolding of these proteins. I would guess more crosslinks may form when proteins unfold because more possible residue pairs are brought into close proximity. Why the authors observe less crosslinks for WT tau at 50°C compared to 25°C?

5. The crosslinking time is inconsistency throughout the paper. In the method part, the authors stated with DSS they did crosslinking for three minutes and for DMTMM was 15 minutes. But in the figure legend of Supplementary figure 2 and 3, one minute crosslinking time was used.

Reviewer #4 (Remarks to the Author):

The manuscript from Hou et al. represents another solid contribution from the Joachimiak lab on characterizing the modulation of amyloid aggregation by chaperones. The multidisciplinary approach taken in this manuscript is an appropriate way to think about the multifaceted effect of chaperones on protein self-assembly. This manuscript is timely since the protecting role of DNAJ proteins against amyloid formation has been increasingly recognised in the field. The role of class A and class B DNAJ proteins has recently been described in the literature, but little is known about class C DNAJs. In this regard, the manuscript may benefit from a comparison between DNAJB1, DNAJB6 and DNAJC7 in the discussion. The MS experiments are well thought through and described, and the manuscript is well

written. However, I have some concerns that need to be addressed.

Major concerns:

1. Much of the paper is geared around the characterization of the interaction between monomeric tau and monomeric DNAJC7. The authors should provide direct evidence for the stoichiometry of the binding reaction. The majority of chaperones that are effective against protein self-assembly do not act on the monomeric level but instead bind aggregated species. Some of these chaperones also oligomerize themselves. In the manuscript there are several lines of evidence that argue against the importance of monomer-monomer binding. The individual TPR domains, although having an 8-10 fold higher affinity for tau than the full length protein, do not reduce aggregation but instead increase it. How is that possible? The results of the in vitro seeding assays are not consistent with the measured monomer-monomer affinities using MST (see point 3). Is multivalency a better model to explain some of these data?
2. One of the main conclusions of the paper is that DNAJC7 recognises a tau conformation in which the N-terminus and the repeat domains are in close contact, with this conformation being less populated in the P301L mutant. This hypothesis is mainly supported by XL-MS data but I feel that a complementary technique would make this conclusion a lot stronger. The authors try to address this by using NMR CSPs shown in Figure 2 but those are very weak and perhaps not compatible with the measured Kds (see point 3). The authors could consider single molecule FRET studies to detect contacts involving the repeat domains and how these are modulated in the presence of DNAJC7 or in the P301L mutant. Maybe NMR studies using spin labels can also help?
3. Although the results of the various experiments agree qualitatively in most cases, they are not quantitatively consistent with each other. For example, the MST measured Kd for the tauRD-DNAJC7 interaction is 2 μ M. Is the magnitude of the observed CSPs in Figure 2 consistent with this affinity? what are the concentrations used for the NMR CSP experiment? Assuming that the model of tau inhibition is that DNAJC7 depletes tau monomers, the elongation rate in the in vitro seeding experiment in Figure 4 should be proportional to [tau monomer]. Upon addition of DNAJC7 the rate of aggregation should follow a binding curve (ie more DNAJC7, more bound tau monomer, less aggregation). This seems to be the case, but the exact numbers do not match with the Kd of 0.5 nM – i.e. 0.1x of DNAJC7 should have very little effect on aggregation and 0.35x should show a ~20% decrease in rate. This problem is even more pronounced for P301L where the 1X curve is essentially the same as WT even though the Kd is 16x higher (see Fig1).

Minor

1. Extreme care should be taken when modelling interactions between flexible proteins using sparse data. I am not sure what conclusions can be drawn from rosetta calculations using a homology model of DNAJC7 and a few crosslinks to dock tau. What does a representative model mean in such systems? Also, how do the calculations in Figure 2g-j show that R1R2 is more exposed?
2. What kind of chemical shifts (N, HN, CA, CB) were used in cs-rosetta? Is there any information about propensity for a local R1R2 structure in these?
3. Please add a figure to show the position of all available lysines on the sequence or structure
4. I found the use of 'natively folded tau' notion a bit confusing since the authors probably refer to an element that has some propensity for structure rather than being natively folded.
4. The authors propose that P301L changes the dynamics of the repeat domain using cross links at different temperatures. This is an intriguing experiment but I find it hard to interpret it as a change in

dynamics.

5. 1.5 μM is mentioned as 0.35X in the caption but 0.5X in Figure 4 legend

6. Y Axis label is missing in Figure 5b

We are very grateful to the reviewers for their critical and detailed evaluation of our study and very helpful suggestions. We also thank the reviewers for their positive assessment of our work. Please find below our point-by-point response to the reviewer's criticisms. We have now carried out a series of new experiments suggested by reviewers including data on DnaJC7 mutants, new data analyses and carried out significant revisions to the manuscript, addressing reviewer comments in full to improve and streamline the final manuscript. We hope that they will find this new version suitable for publication in Nat Comm.

REVIEWER COMMENTS

Reviewer #1 (Remarks to the Author):

In this work, the authors explore differential binding of proteins to tau during its maturation as an aggregation seed. Mass spectrometry analysis of these samples suggested that a number of chaperones, including Hsp40s, interact with tau. The authors focus on those that change association with tau during this time, leading them to DnaJC7 (although DnaJC5 is also quite interesting). They then proceed to study this interaction in more detail, suggesting that DnaJC7 binds the microtubule-binding repeats R1 and R2 (Fig 2), prefers the WT state (Fig 4) and that the interaction may occur through all three TPR domains (Fig 5). Perhaps most strikingly, they show that CRIPSR knockdown of DnaJC7 increases seeding (Fig 1C). Although not a major focus of the work, the observed difference in crosslinks between WT and P301L full length tau proteins (in the absence of DnaJC7) is a nice addition (fig 3c, e), as it extends from the author's previous work on shorter, RD versions. Overall, this is an interesting and important work that suggests a role for DnaJC7 in tau aggregation. The analysis of tau-associated protein during seeding in Figure 1 and the Supplemental is also likely of broad interest. However, some of the interpretations seem premature and there needs to be more rigorous controls in places.

Major:

1. An alternative, simple explanation for the difference in DnaJC7's affinity between full length and RD tau is that there are additional, low affinity binding sites for the chaperone in the N- and C-terminal regions, adding up to a tighter apparent affinity. What is the stoichiometry of the DnaJC7-tau complex in both cases? Hill slope from the MST? Stoichiometry by SEC-MALS? Chaperones are notoriously "sticky" so it would not be surprising to find secondary sites. This is an important question because another simple reason for the change in apparent affinity between WT and P301L is a change in stoichiometry. (i.e. distinct from the current interpretation of a change in tau "structure").

We thank the reviewer for this comment. Usually stoichiometry measurements like ultracentrifuge analysis require long data acquisition times (20hrs) at high concentrations of protein, which is not ideal for our system. Instead, we used established approaches to reinterpret Microscale Thermophoresis (MST) binding data to evaluate WT tau: WT DnaJC7 and P301L tau: WT DnaJC7 binding stoichiometry. We

used comparison of T_{jump} (T_j) and $T_{\text{thermophoresis}}$ (T_m). The MST not only gives binding constants but also can be used to determine interaction stoichiometries [1]. An example of 1:1 and 2:1 discrepancy between T_j and T_m curve from the literature [1] is shown in the response Figure 1a and b below. When a single binding mode exists, the T_j describes the rapid change surrounding the fluorophore within a second and should be similar to $T_{\text{thermophoresis}}$ (T_m) which describes the global complex properties on a second to minute scale (see below, response Fig. 1a). When a different stoichiometry exists in the reaction, there will be deviation in the shape of the curves comparing the T_j and T_m experiments (see below, response Fig. 1b). In contrast, in our binding data involving DnaJC7, the T_j and T_m curves of both DnaJC7 binding to WT and P301L tau is matching the 1:1 binding mode (see below, response Fig. 1c and d). Based on the existing shape of the curves (ie typical sigmoidal shape) the stoichiometry should be 1:1 as T_j and T_m curves are identical suggesting that the initial binding events are identical to the equilibrated events. This suggests that there are no obvious secondary binding sites for DnaJC7 binding to WT tau or P301L tau.

Response Figure 1. Analysis of MST data shows DnaJC7 binds to WT and P301L tau in a 1:1 stoichiometry. a. Titration of 1:1 stoichiometry binding model shows a typical, sigmoidal shape between T_j (left) and T_m (right). b. Titration of 2:1 stoichiometry binding model shows striking difference between T_j (left) and T_m (right). c. T_j from WT

tau:DnaJC7 titration curve equivalent to T_m . d. T_j from P301L tau:DnaJC7 equivalent to T_m .

Additionally, our crosslinking experiments on WT tau:DnaJC7 and P301L tau:DnaJC7 using DSS (revised supplementary Fig 3b) but also using a non specific crosslinker, glutaraldehyde (see below, response Fig. 2), does not show formation of species larger than a heterodimer consistent with a 1:1 stoichiometry. Together our new data suggest that the binding mode between WT tau and DnaJC7 is 1:1 and that the decrease in P301L tau binding to DnaJC7 is not due to changes in stoichiometry.

GA crosslinking experiment

Response Figure 2. non-specific crosslinking of DnaJC7 to WT (left) and P301L (right) tau using glutaraldehyde acid (GA) shows only hetero-dimer formed. Hetero-dimer is indicated by an arrow.

2. In the crosslinking experiments with DnaJC7 (Fig 3), it is not clear if the chaperone is present at its K_d concentration, as the two tau constructs have different affinity values? Alternatively, if the concentration of DnaJC7 is saturating, then why are there differences observed? This experiment needs to be designed to allow thermodynamic comparisons between the two samples and it was difficult to parse the required details from the legends or methods.

The reviewer brings up a great point. In the crosslinking experiments, we used similar binding conditions for WT tau and P301L tau with DnaJC7. In brief, 17.6 μM WT or P301L tau was mixing with 25mM BME and boiled at 100°C for 5 minutes, then 1:1 (vol/vol) diluted with 1XPBS to 8.8 μM with 25 μM Thioflavin added. For each well measured, 30 μl WT tau or P301L tau was mixed with 15 μl seeding monomer and 15 μl titrated DNAJC7 with final concentration ranging from 0.1 μM to 4.4 μM . The

fluorescence intensities are indeed different between WT and P301L tau suggesting that there is a difference in steady state binding given these conditions. Importantly, we extract heterodimer bands from the gel, process the samples and for the mass spectrometry analysis, inject 1ug of purified crosslinked peptides for each experiment. Despite differences in affinity, in the mass spec experiments we are comparing equal amounts of peptides for each sample allowing a direct comparison of the peptide crosslink patterns between WT tau:DnaJC7 and P301Ltau:DnaJC7 (see XL-MS mass spectrometry intensity values in Supplementary Data 2). Additionally, we observe similar amounts of crosslinks (and monolinks/looplinks) across the conditions further supporting a direct comparison of the two datasets (Fig. 3). We have clarified this description in the results (Lines 235-237) and in the methods on (Lines 733).

3. Related to this comment, is there sufficient time in the crosslinking experiments to allow for saturable DnaJC7 binding before the crosslinking reactions? As above, there needs to be a careful examination of kinetic parameters.

Thank you for this question. Related to the above description, we optimized variables (protein ratios, time and crosslinkers) in the crosslinking conditions to produce reproducible formation of bands corresponding to heterodimers between tau and DnaJC7. For example, using DMTMM or DMTMM/ADH crosslinkers we did not detect formation of heterodimer bands. For DSS reactions, we ascertained that 1:1.2 ratios of tau and DnaJC7 followed by 1 hour incubation at RT and then followed by a 1min DSS reaction at 37°C yielded consistent appearance of a hetero-dimer band across protein preps. In follow up studies on this project, we are planning to employ binding measurements that will determine kinetics of binding and how the k_{on} and k_{off} rates partition between the WT tau and P301L tau binding to DnaJC7. We have clarified the description of the reaction conditions in the methods (Lines 714-717)

4. More work is needed to ensure that DnaJC7 TPR truncations (TPR1 20-156, etc) are folded. It seems that the results could be explained by differences in the solubility, aggregation or folding of the truncations. Are they folded, as judged by circular dichroism? Are they monomeric on SEC (esp at the concentrations and buffers and times and temperatures employed)? This question arises, in part, from the counter-intuitive and somewhat confusing observation that some of the truncations have a tighter apparent affinity than the full length. There is either an auto-inhibitory mechanism (which would be unexpected amongst the TPR proteins) or, more likely, some of the constructs are (at least partially) aggregated.

We thank the reviewer for this insightful question. From the experiments we had no indications that the TPR domains were aggregated or not folded as the crosslinking did not yield formation of larger species very common if proteins are aggregated or misfolded. To further confirm that these single TPR domain proteins are folded properly we directly compared each TPR domain by SEC and show that they are monomeric and elute around the size of Ribonuclease (13,700Da, see below, response Fig. 3). Additionally, we also performed DLS on the domains and show that there are no aggregates (consistent with SEC) and the R_h remains small (see below, response Fig.

5). We have also repeated the DSS-based XL-MS experiments between the TPR domains and WT tau and P301L tau (response Fig 4 and revised supplementary Fig 5) and again observe no clear evidence of the individual domains aggregating in the presence of crosslinker and furthermore the reactions containing individual TPR domains and tau yield formation of a clear hetero-dimer band. Finally, the TPR2b:tau crosslinking experiments reveal similar contacts as observed between tau and TPR2b in full-length DnaJC7 supporting that this domain is not only folded but can recapitulate binding modes observed in the full-length protein. This observation is consistent with the TPR2b domain alone binding to tau with the highest affinity. We indeed do not yet fully understand why this domain alone can bind to tau (recapitulating contacts observed in FL) but is unable to regulate aggregation. This is an interesting observation and suggests that simply binding is not sufficient to impart aggregation suppression activity. Indeed, there are examples of antibodies binding to amyloidogenic proteins that have no effect on the substrate aggregation. This also highlights that simply binding a small domain to an epitope is not enough to prevent incorporation of that protein into assemblies especially since this R1R2 epitope is upstream of the core elements including the VQIVYK amyloid motif involved in fibrillization. We are really interested in understanding how DnaJC7 binds to full-length tau using cryo-EM but also how DnaJC7 binds to short epitopes. These experiments are ongoing and will be a part of future manuscripts.

Response Figure 3. SEC profiles of three TPR domains show a molecular weight close to elution position of Ribonuclease A at around 13.7KDa.

Response Figure 4. DSS crosslinking of individual TPR domains (TPR1, TPR2a and TPR2b) preincubated with tau exhibits a hetero-dimer band (red box).

Response Figure 5. Radius of hydration from DLS for TPR1 (left), TPR2a (middle) and TPR2b are consistent with Rh value derived from a structural model of the isolated TPR domain.

5. Related to this comment, the truncation of TPR domains doesn't seem like the best way to address the question of where the tau interaction is taking place. With the modelling (Fig 6), point mutations could be made in the context of full length DnaJC7 to test this idea more specifically.

This is a fantastic question and motivated by the reviewer's suggestion we decided to test our structural model directly with mutagenesis. Informed by our crosslinking (both tauRD and FL tau) experiments, we designed a DnaJC7 mutant that modified 6 residues on the first helix of TPR2b that fall roughly between the K254 and K306 crosslink sites (revised Fig. 6ab). We termed this mutant DnaJC7_{TPR2b_mut}. The DnaJC7_{TPR2b_mut} mutant was purified using Ni-NTA, SP ion exchange and SEC and showed a nice symmetric peak corresponding to a monomer. We first performed MST binding experiments to measure binding to WT tau. We again observe that DnaJC7 binds to WT tau with 0.35uM affinity but the DnaJC7_{TPR2b_mut} binds with 6uM affinity (revised Fig. 6c) suggesting a 20-fold reduction in affinity! We followed these

experiments with in vitro ThT aggregation assays followed by TEM and seeding experiments to further quantify the aggregates in the samples at the end of the assay. Consistent with the decreased binding affinity we observe a delay in aggregation and the final amplitude of the ThT fluorescence is also reduced (revised Fig. 6d). We are not sure why the shape of the curve is different but we suspect that there may be other secondary binding sites that may contribute to DnaJC7 activity (consistent with other TPRs weakly contributing to binding). TEM images of the endpoints is consistent, WT DnaJC7 can completely abolish aggregation of WT tau while DnaJC7_{TPR2b_mut} is less efficient allowing us to still find tau fibrils by TEM (revised supplementary Fig. 6a). Finally, the seeding data on these samples suggests under addition of DnaJC7_{TPR2b_mut} into the aggregation reaction, the partial loss of DnaJC7 function leads to enhanced seeding activity (revised Fig. 6e). These exciting data independently validate that our XL-MS approach identifies bona fide binding sites which are later confirmed using modeling (revised Fig. 7).

6. The observation that P270S tauRD does not bind DnaJC7 is interesting, but it is not clear whether it makes thermodynamic sense? What is the energetic difference between the folded and unfolded state? It seems unlikely to be very large, given that the protein is an IDP. So, wouldn't the P270S sample both states – and therefore a portion of the protein be accessible for binding DnaJC7? Tau has few residues for easy protein concentration determination (e.g. Trp), so one wonders if the concentrations of the two samples are just different? Or P270S tau RD is mildly aggregation prone under these conditions and not as available for binding? The interpretation of this part of the manuscript (Fig 6f, g) seems speculative.

We thank the reviewer for this observation. In our manuscript, we do not claim that the P270S mutation unfolds tau or even makes tau more aggregation prone. Our data does support that P270S tauRD has similar aggregation propensity to WT tauRD so its availability for binding should be comparable to WT tauRD (revised Supplementary Fig 7b). In prior work, we have shown that the R1R2 peptide does not aggregate alone and incorporation of the P270S mutation does not change this behavior (Chen et al Nat Comm 2019). In this study we show that despite similar aggregation propensities to WT tauRD, the P270S tauRD does not bind well to DnaJC7 (revised Supplementary Fig 7a) which is also consistent with NMR CSPs and broadening data in support that the R1R2 element in tau including the PGGG motif (see revised Supplementary Fig. 2e) are involved in binding DnaJC7. Additionally, from the ensemble docking experiments in this study (revised Fig 7), we predicted that the R1R2 peptide could bind to DnaJC7 in a “collapsed” conformation while “expanded” conformations had poorer binding energies. From these data, we hypothesized that the WT R1R2 epitope could bind to DnaJC7 and interfere with tau binding while a P270S mutant would be less efficient at binding to DnaJC7 and thus should also not interfere with binding. We are very excited about these results and we have ongoing structural studies to determine structures of DnaJC7 bound to R1R2 for which we are currently optimizing crystals and are coordinating time to screen

7. Does DnaJC7 block the aggregation of P301L tau seeded with WT aggregates (as in Figure 4)? That experiment would discern whether the difference in activity of DnaJC7 between the WT and P301L was due to chaperone binding to the monomer or the seed. The uniform lag time in the ThT assays (see Fig 4A, B) and the equimolar requirement suggest that the binding activity is on monomer, but the mixed seeding experiment would finalize that idea.

Great question which turned out to motivate this work! In revised figure 4, full-length WT tau and P301L tau were seeded with soluble monomer/dimer tau seeds isolated from full-length WT tau treated with heparin (Mirbaha et al eLife 2018 and more recently Hou et al Sci Rep 2021). We have attempted to make seeding monomer using FL 2N4R P301L tau but because the pathogenic mutation makes the seeds more aggregation-prone it is really difficult to isolate intermediates including monomer to use in seeding experiments. In our initial binding experiments, we explicitly measured binding affinity of DnaJC7 for monomeric forms of WT tau, P301L tau or the WT monomer tau seed (see revised Supplementary Figure 4a). These data suggest that DnaJC7 binds these species in the following order WT tau (400 nM) > P301L tau (8 uM) > tau seed (30 uM). Given that the amount of seed used in the aggregation reactions is in the low nanomolar concentration, our determined affinities suggest that DnaJC7 impacts the aggregation process by binding to the monomer and not the soluble tau seeds.

8. Throughout the figures, the individual data points for replicates should be shown to align with Nature/Springer standards. This is especially important for some of the more subtle differences (Fig 2b, Fig 4C, D). Also, it needs to be clear if replicates are independent (e.g different batches of protein) or technical.

We apologize that the plots in Fig 2b, Fig 4c,d do not adhere to the Nature springer standards. These figures have been corrected to include individual data points for the replicates. For Fig 2b we have included a plot in Supplementary Fig. 2a highlighting the individual points from three replicate experiments. We have also clarified in the legends and in the methods throughout whether the experiments were performed using technical or biological replicates.

Minor:

1. The argument that the MD results align with the crosslinking (page 16, line 400) is circular, as the crosslinking was used to guide/restrict the docking.

We thank the reviewer for raising this concern. We try to be careful with how the crosslinks are interpreted in the context of models and typically avoid explicitly constraining the simulations using the crosslink data and thus most often carry out unrestrained docking first at low resolution and then followed by high resolution. In this paper, we used a new docking procedure that incorporates an ensemble of peptide conformations (different conformations that include both collapsed (ie hairpin) but also expanded) against DnaJC7. For each unrestrained model of the DnaJC7:R1R2 (or

R2R3) complex we calculated the predicted binding energy. We then mapped the intermolecular crosslinks and compared them to the predicted binding energies. Using this unbiased strategy, we identified models of complexes that had nearly the lowest energetics in the entire ensemble and were consistent with the experimental intermolecular crosslinks between tau and DnaJC7. Thus the “hairpin” conformation of the peptide was selected purely based on energy and not crosslinks but in the end was consistent with intramolecular crosslink data. This model also explained the experimental intramolecular crosslinks within tau – these were also not used explicitly as a criteria to select the models. This is clarified in Lines 793-799

2. The sentence on page 4, line 66-68 seems redundant with others in the paragraph.

We have removed this sentence as suggested by the reviewer.

3. Hsp40s is no longer the preferred nomenclature for the J-domain proteins (Kampinga et al. 2019 Cell Stress Chaperones), largely because members of this class of proteins have molecular mass values that are not uniformly 40 kDa. For example, DnaJC7 is ~55 kDa.

We have gone through the entire manuscript and have replaced instances of Hsp40 with J-domain protein. We thank the reviewer for this suggestion.

4. The use of “dramatically” and “high affinity” in page 12 (line 291 and 296, respectively) don’t seem justified by the 2-fold changes in crosslinking and the high nanomolar affinity constants.

Thank you -- we have reworded our statements.

5. How many significant figures are appropriate for the cell-based seeding assay? Currently, the results are reported as three (69.7 ± 5.3 %), but given the error, it seems that two might be more appropriate (e.g. 70 ± 5.3 %).

As suggested by the reviewer we have reduced the number of significant digits reported for the seeding assay.

Reviewer #2 (Remarks to the Author):

This is an interesting work that reports the interaction between DnaJC7 and tau. The authors extended their previous work on tau and here they used a combined approach to investigate the binding of tau with DnaJC7 and how the chaperone inhibits tau's aggregation. They found out that several chaperones, including an number of DnaJs (Hsp40s) engage tau. They decided to focus on DnaJC7 given that no substrates have been reported for this protein. Cross-linking Mass Spectrometry was employed to probe for the interaction, complemented with NMR, binding assays and finally coarse-grained docking. This is arguably a challenging system and thus a high-resolution structure of

the complex cannot be easily obtained. Nevertheless, the insight provided by this work will be of great interest to the community that justifies publication at Nat Comm. It is particularly intriguing that the unfolded variant of tau appears to bind with lower affinity relative to the "more folded" wild type tau. The experiments are very well executed, clearly described and the findings efficiently presented in the text and in the figures. I only have some minor comments that hopefully can provide some clarity and improve the manuscript.

1. Do the authors have any data regarding the binding of the N-terminal region of tau to DnaJC7? Do they know where on DnaJC7 it binds?

We thank the reviewer for this insightful question. Full-length WT tau binds to DnaJC7 with 400nM affinity. We have measured the binding affinity between the N-terminus of tau (residues 1-242) and DnaJC7 and determined an affinity of $17 \pm 4 \mu\text{M}$ (supplementary Fig. 3a). These data suggest that this region alone does not bind to DnaJC7 efficiently – a little weaker than the repeat domain of tau alone. Thus our data support that there is synergy between the N-terminus and repeat domain binding to DnaJC7. In our XL-MS experiments derived from FL WT tau:DnaJC7 we predominantly observe crosslinks from the TPR2b domain of DnaJC7 to the pro-rich domain and the repeat domains – these cumulative contacts determine the 400nM affinity. In addition to these contacts we also observe some isolated contacts between tau and the other TPR domains (TPR1 and TPR2a) but these have much lower frequencies and thus are also less important.

2. It is surprising that the J domain seems to bind to tau, albeit with low affinity. Still an 8 microM affinity is not trivial. Is this within experimental noise or the authors believe there is indeed binding?

We were indeed surprised by this result. While we agree that the 8uM affinity observed by the JD alone is not trivial it is definitely much weaker than the other TPR domains. The JD encodes some surface charges (see revised Figure 5b) and this affinity might be driven by these properties but we do not think this domain contributes much to the overall binding between tau and DnaJC7 not aggregation suppression activity.

3. In the NMR titration experiment of labeled tau by DnaJC7, several resonances broaden beyond detection. What are these residues? And what is the reason for experiencing line broadening? Is it because of their direct interaction with DnaJC7? In this case, wouldn't this be suggestive of a stronger binding compared to the other residues that simply display chemical shift perturbation? Is there assignment for these residues?

We thank the reviewer for this key observation. We have previously looked at broadening of the intensities in the spectra comparing the peak signal intensity for ^{15}N tauRD alone and in the presence of DnaJC7. We observed overall broadening and interpreted this as binding of the larger DnaJC7 (55kDa) to the smaller ^{15}N tauRD (15kDa) which makes the interpretation of the broadening more complicated and noisy.

As the reviewer astutely observed in our spectra, there are peaks that broaden beyond detection as DnaJC7 is titrated (see response Fig. 6 below, S258, S262, T263, N265 and K274) that cluster to the R1R2 element. Additionally, we observe broadening of peaks assigned as glycine. While we do not have specific amino acid position assignments for some of these glycine residues because of sequence degeneracy across the four repeats including the PGGG motifs which lie at the interfaces between the repeats. The broadening of the signal at these sites (see response Fig. 6 below, G271/G302) would also be consistent with binding to the PGGG turn in the R1R2 (possible also R2R3). Together, in these NMR experiments we observe a combination of CSPs and peak broadening that localizes to the R1R2 including the PGGG turn suggesting that DnaJC7 recognizes the R1R2 element using slow and fast binding kinetics. We have included a discussion of the above peaks that broaden in response to DnaJC7 binding in the results on lines 189-195 and included the peak broadening analysis in supplementary Fig. 2e.

Response Figure 6. Peak broadening in HSQC ^{15}N - ^1H NMR for tauRD with DnaJC7. Example of HSQC ^{15}N - ^1H NMR peaks located in the R1R2 domain that broaden with increasing concentration of titrated DnaJC7. Peak broadening comparing intensities of peaks from ^{15}N tauRD with 1X DnaJC7 (I) vs ^{15}N tauRD alone (I_0). Residues from inset that broaden are highlighted.

4. The authors state that there is very little information about how Hsp40s recognize and interact with substrates. I am thus surprised that the authors did not mention or discuss a key structure determined recently of a full-length Hsp40 in complex with an unfolded substrate (PMID 31604242). It would be of interest if the authors compared the mode of binding reported here with the structure of the Hsp40 reported previously, even if the

binding domains are different.

As suggested by the reviewer we have compared the binding mode of DnaJC7:tau and the ttHsp40:phoQ complex. While the binding is quite different given the unique architecture of DnaJC7 compared to the more classical dimeric Hsp40, we observe that DnaJC7 binds to tau with high affinity (unique to chaperones!) and recognizes a short specific motif within the repeat domain in a native-like conformation. In contrast, in Jiang et al, the authors used high-resolution NMR to study the substrate interaction with a canonical JDP from *Thermus Thermophilus* and observe that the JPDs dynamically binds non-polar elements in the unfolded substrate through multi-valent interactions likely suggesting a possible mode for prevention of aggregation. We have included a brief comparison of the DnaJC7:tau interaction compared to ttJPD:phoQ (Line 499-503).

Reviewer #3 (Remarks to the Author):

The manuscript from Hou et al nicely illustrated the role of DnaJC7 in stabilizing natively folded tau to prevent tau conversion. However, I still have a few concerns, especially to the data quality of the quantitative crosslinking analysis. My points are as follows.

1. My biggest concern is related to the crosslink comparison between different species, i.e. WT tau, P301L tau, WT tau+DnaJC7 and P301L tau+DnaJC7. I am puzzled that all analyses were done based on crosslink numbers, rather than standard quantitative MS analysis. Given that the authors acquired many replicates per sample, it would be much more accurate to use label free quantitative pipeline rather than merely counting crosslink numbers. Therefore I would suggest the authors to re-analyze their crosslinking data using label free quantitation.

We apologize that we were not clear about our XL-MS analysis approach to interpret the replicate crosslink data sets. We also want to clarify that we are only using the crosslink counts from the consensus datasets to summarize the data in an easily digestible format (i.e. bar plots in Fig. 3b,d,g). In particular, this is useful when we compare changes in the distribution of identified consensus contacts in the different regions of the protein. As an example, the comparison of the crosslink distributions are important and highlight changes in crosslinks between samples (i.e. Fig 3e) where in WT we observe many contacts to K311 but these are absent in the P301L mutant. This observation is consistent with our prior work on the tau repeat domain using a similar analysis (Chen et al 2019). We had in the past developed a quantitative tool in collaboration with the Aebersold lab for XL-MS data analysis (xtract; <https://www.nature.com/articles/nmeth.3631>) to help quantify differences across samples. In that study we used TRiC/CCT and Luciferase as model systems to test this new tool. One large advantage was the reproducible patterns of the detected contacts in these systems because the proteins are folded and stable. Thus we were able to cleanly measure changes in the contacts due to a perturbation allowing the calculation of enrichment per pair of contacts between datasets. In the IDP systems that we study (i.e. tau) there is a lot more heterogeneity in the XL-MS data. As an example, the above K311-based set of contacts are only present in one consensus dataset (WT) but not in

the other (P301L). To overcome this heterogeneity we devised the consensus mapping approach from replicates to gain insight into contacts in IDPs that are more probable and so far it has proven very powerful. Additionally, we have employed temperature as perturbations and the combination of consensus datasets and temperature has also been very insightful (based on prior work, Chen et al 2019 Nat Comm). I definitely agree that we should begin incorporating quantitative analysis into our XL-MS IDP experiments but we would need to use the appropriate system (tauRD for example would be more ideal compared to FL tau) this will likely take additional method development before it can be suitably implemented in our systems.

To help clarify our approach to interpreting the data we have revised the methods section (Lines 751-757) to explain in detail how we employ the consensus datasets allowing interpretation of changes in conformationally heterogeneous samples. Additionally, to help determine that the individual datasets are comparable we have included in Supplementary Data 2 the max intensities for all datasets analyzed in this study.

2. The authors extracted the gel bands corresponding to the species of interest for crosslinking analysis. Did they make sure that the same amount of peptides were loaded on LCMS, or were some kind of normalization method applied between samples? Since only crosslink counts were considered for quantification and the number of crosslinks vary significantly depending on the sample load, I am skeptical about how accurate the quantification is in the whole crosslinking analysis. As shown in Supplementary 3b, the intensities of Coomassie staining are rather different in different samples. Especially for the DNAJC7+tau heterodimer, I could barely see any gel bands at the corresponding molecular weight.

This is a great point and is essential for our ability to compare the datasets across samples. Our strategy to extract bands from the gels is essential to allow comparison of samples especially given differences in affinity between DnaJC7 and the different forms of tau. Crosslinked reactions of tau and chaperone were resolved by SDS-PAGE and the hetero-dimer band was excised (see Supplementary Fig. 3b and 5a, hetero-dimer bands denoted with an arrow) which was then used for in gel digestion and subsequent processing. For each condition, we injected 1ug of purified crosslinked peptides in each LC-MS run ensuring that we will be able to compare across conditions. Additionally, we ensured that the peak intensities for each sample are comparable, the max intensities for all the experiments are reported in the revised Supplementary Data 2.

3. In general, I found the crosslink plots shown as supplementary figure c-e are difficult to read. Surely I can see the differences in the boxed areas (for instance the dashed boxes between WT and P301L tau) but I also see changes in other regions of the plot. I think the authors should provide either a cutoff value or some statistical measurements to distinguish significant from non-significant changes. Another point is in supplementary figure e, the authors claim a complete absence of long range contacts

with the N-terminus of WT tau (from WT tau+DnaJC7 complex), but I do see a dot connecting residues between 0-50 to 350-400. Am I taking something wrong here?

We thank the reviewer for raising this concern. As tau is an IDP, our XL-MS method yields only medium to low resolution information on the overall dynamics of the molecules. Hence, the global changes of tau conformational dynamics established by the contact maps is complicated but yet we see defined changes in patterns between the WT and the P301L tau mutant when utilizing our established consensus crosslink mapping strategy. In order to make the findings statistically significant, we use established procedures in our field to first curate each replicate dataset with a defined score cutoff in Xquest yielding low FDRs for each independent dataset. For folded proteins, single datasets are used to make conclusions for XL-MS analysis. For IDPs, however, we have found that assembling a consensus mapping across replicates yields highly reproducible datasets that allow inference of more subtle changes in conformation despite the complexity of the data (Mirbaha et al 2018 elife, Chen et al 2019 Nat Comm, Hou et al 2021 Sci Rep). In this manuscript, we primarily focus on the most dramatic differences between WT and P301L tau, such as the contacts that involve K311. This observation is consistent with our prior XL-MS experiments comparing WT tauRD and P301L tauRD which we previously proposed as changes in “stability” by acquiring data across temperatures. Another notable difference between WT and P301L tau involves the interactions with the N-term which are a little more subtle than the K311 but nonetheless noteworthy. Derived from our data these two features appear to differentiate WT and P301L tau. For other parts of the protein, it is not surprising that the other region of the dataset has less obvious differences/variation and the physiological significance of these changes will require more investigation in future including possible quantitative approaches such as Xtract. As our intention was to focus on the most obvious differences because the complexity of the data, this is why we obtained to highlight the largest differences to emphasize loss of crosslink pairs in the consensus mapping datasets which we propose alter the dynamics of tau via both global and local conformational changes which are consistent with our prior studies indicating the P301L mutation in tau promote unfolding and thus result in unfolding of tau. This kind of mechanism for formation of aggregation-prone species is supported by prior work to highlight increases in exposure of amyloid motifs that can mediate pathogenicity of a tau molecule (Mirbaha et al 2018 elife, Chen et al 2019 Nat Comm, Hou et al 2021 Sci Rep). This is also precisely why we decided to summarize the data in bar plots in the main figure – the boxed areas highlighted in the supplementary figures are shown as bar plots in the main figure. The plots in supplementary figure 3 c-e are indeed quite complicated even though they are an attempt to summarize the consensus patterns observed across 5 replicate datasets. Regarding the last question: “Another point is in supplementary figure e, the authors claim a complete absence of long range contacts with the N-terminus of WT tau (from WT tau+DnaJC7 complex), but I do see a dot connecting residues between 0-50 to 350-400.” We apologize for this statement and realized that the plots for these data were not updated. We have included a revised panel for Supplementary Figure 3e and have rephrased our statement that there is a large reduction in the crosslinks on lines 289-291.

4. What is the reason that crosslinking was performed at 50 °C? The authors stated in the paper “we employed these two temperatures to probe the stability of the intensities in response to unfolding”, but I didn’t find any data supporting how temperature affects unfolding of these proteins. I would guess more crosslinks may form when proteins unfold because more possible residue pairs are brought into close proximity. Why the authors observe less crosslinks for WT tau at 50°C compared to 25°C?

This is a great question and we apologize that we did not explain the motivation behind our approach to collect data at several temperatures. In our previous work on tau we have employed a temperature gradient coupled to XL-MS to follow unfolding of tau with increasing temperature. This work revealed a possible mechanism for how pathogenic mutations in tau can modify local conformation which lead to subsequent protein aggregation (Chen et al Nat Comm 2019). This approach has now been widely cited and expounded on in reviews including an invited chapter in *Methods in Mol Biology* that is *in press*. We believe that these comparisons are extremely valuable and allow more precise observation of changes in protein structure conformation in response to perturbations. In addition, to following crosslink patterns we have begun incorporating frequencies of monolinks and loop links which report on solvent exposed regions. The combined crosslink, monolink/loop link analysis with temperature modifications is an important step in XL-MS methods to yield important biological insight into conformational changes in intrinsically disordered proteins where the structural elements features are more subtle than in globular proteins. Indeed, similar attempts have been incorporated into NMR experiments that revealed subtle changes in local structure in abeta [2].

5. The crosslinking time is inconsistency throughout the paper. In the method part, the authors stated with DSS they did crosslinking for three minutes and for DMTMM was 15 minutes. But in the figure legend of Supplementary figure 2 and 3, one minute crosslinking time was used.

We thank the reviewer for catching this typo. DSS reactions were carried out for 1 min and DMTMM reactions were carried for 15 minutes. We have fixed the details of the method description.

Reviewer #4 (Remarks to the Author):

The manuscript from Hou et al. represents another solid contribution from the Joachimiak lab on characterizing the modulation of amyloid aggregation by chaperones. The multidisciplinary approach taken in this manuscript is an appropriate way to think about the multifaceted effect of chaperones on protein self-assembly. This manuscript is timely since the protecting role of DNAJ proteins against amyloid formation has been increasingly recognised in the field. The role of class A and class B DNAJ proteins has recently been described in the literature, but little is known about class C DNAJs. In this regard, the manuscript may benefit from a comparison between DNAJB1, DNAJB6 and

DNAJC7 in the discussion. The MS experiments are well thought through and described, and the manuscript is well written. However, I have some concerns that need to be addressed.

Major concerns:

1. Much of the paper is geared around the characterization of the interaction between monomeric tau and monomeric DNAJC7. The authors should provide direct evidence for the stoichiometry of the binding reaction. The majority of chaperones that are effective against protein self-assembly do not act on the monomeric level but instead bind aggregated species. Some of these chaperones also oligomerize themselves. In the manuscript there are several lines of evidence that argue against the importance of monomer-monomer binding. The individual TPR domains, although having an 8-10 fold higher affinity for tau than the full length protein, do not reduce aggregation but instead increase it. How is that possible? The results of the in vitro seeding assays are not consistent with the measured monomer-monomer affinities using MST (see point 3). Is multivalency a better model to explain some of these data?

We acknowledge the reviewer's point which was also raised by Reviewer #1. In terms of domain architecture DnaJC7 is unique compared to classical Hsp40s (DnaJB1 or DnaJA2) or other chaperones previously determined to bind to amyloidogenic substrates. Indeed small heat shock proteins (HSPB family) and a subset of B' family Hsp40s (DnaJB6 or DnaJB8) can form oligomeric assemblies but it remains unclear how they interact with substrates and whether they bind to monomers, pathogenic seeds, oligomers or fibrils. Data from our lab on DnaJB8 (homolog of DnaJB6) suggests that the chaperone can oligomerize but it is the small species (dimers) that bind to monomers of substrate and prevent their assembly into oligomers. Very recent data on DnaJB1 has shown that it has a preference for binding FL tau:heparin complexes but that it also preferentially binds to monomeric FL P301L tau over monomeric FL WT tau (<https://www.biorxiv.org/content/10.1101/2021.04.11.439324v1>). For DnaJC7, our work supports that the opposite is true and that it has higher affinity to FL WT tau compared to more unfolded pathogenic tau mutants (P301L) or tau seeds. The effect of WT DnaJC7 on WT tau and P301L tau aggregation is also difficult to extrapolate because WT tau and P310L tau have different aggregation propensities so while we can maybe interpret what happens at t=0 based on affinities to the individual species, as the experiment continues the concentrations of different species change which we think overall minimizes the apparent difference in the aggregation kinetics relative to the binding data. Additional unpublished data on DnaJC7 shows that knockout in tau biosensor cells of DnaJC7 increases seeding capacity independent of the seed type, which is not true of DnaJB6 where seeding changes vary with seed source. Reanalysis of the MST data suggests that the binding mode is 1:1 and this is supported by GA crosslinking of the complexes. Similarly, the individual TPR domains also bind in a 1:1 stoichiometry (see response to reviewer #1). We agree and were initially puzzled at the observed affinities of the individual domains and their inability to limit aggregation – TPR2a and TPR2b do not change aggregation but TPR1 increases it slightly. The most important observation/point, however, is that TPR2b binds with the highest affinity and

crosslinking of TPR2b:tau complexes reveals nearly identical contacts observed between TPR2b in full length DnaJC7 suggesting that TPR2b alone encodes specificity for tau binding but the domain alone cannot prevent incorporation of the bound tau to growing tau aggregates (revised Fig 5). Thus this suggests that the other domains may function to sterically prevent the bound tau from being incorporated into an aggregate. To further explain how TPR2b interacts with tau, we designed a DnaJC7 mutant focusing on a helical element guided by the XL-MS data (DnaJC7_{TPR2b_mut}, see revised Fig. 6 and response to reviewer #1 above). The mutant has significantly reduced affinity for tau and is unable to efficiently suppress tau aggregation. Perhaps this is why this chaperone is so exciting because it can extremely efficiently sequester tau from incorporation into tau aggregates – describing a new mechanism for tau aggregation regulation by a chaperone. We have modified our results of data presented in revised Figure 5 but also in the discussion to highlight how we think the cumulative TPR domains work.

2. One of the main conclusions of the paper is that DNAJC7 recognises a tau conformation in which the N-terminus and the repeat domains are in close contact, with this conformation being less populated in the P301L mutant. This hypothesis is mainly supported by XL-MS data but I feel that a complementary technique would make this conclusion a lot stronger. The authors try to address this by using NMR CSPs shown in Figure 2 but those are very weak and perhaps not compatible with the measured Kds (see point 3). The authors could consider single molecule FRET studies to detect contacts involving the repeat domains and how these are modulated in the presence of DNAJC7 or in the P301L mutant. Maybe NMR studies using spin labels can also help?

This is a very interesting point that we are pursuing in future studies in particular to understand how DnaJC7 interacts with different isoforms of tau (0N24 vs 1N2R vs 2N4R) using cryoEM but also using x-ray crystallography for DnaJC7 bound to tau peptides (have optimized crystallization conditions!). There is evidence that the N-domains in tau limit aggregation of tau by interacting with the repeat domain (mandelkow et al and Rhoades et al). In our current observations in the WT tau construct we observe contacts between the acidic N-domains (2N) and the basic repeat domain while in the P301L tau which is more aggregation prone (Chen et al 2019 Nat Comm) we observe that these contacts are more heterogeneous – these conformational changes underlie the ability of the mutant to aggregate faster and are consistent with previously published data on the role of the N-terminus of tau. PRE NMR experiments with full length tau would be extremely interesting but the chemical shifts are not publically available for full-length tau. smFRET experiments could help complement our XL-MS experiments they themselves also may present artifacts based on introduction of mutations in the sequence and addition of large charged/non-polar probes that can change properties of the protein.

3. Although the results of the various experiments agree qualitatively in most cases, they are not quantitatively consistent with each other. For example, the MST measured Kd for the tauRD-DNAJC7 interaction is 2 uM. Is the magnitude of the observed CSPs

in Figure 2 consistent with this affinity? what are the concentrations used for the NMR CSP experiment? Assuming that the model of tau inhibition is that DNAJC7 depletes tau monomers, the elongation rate in the in vitro seeding experiment in Figure 4 should be proportional to [tau monomer]. Upon addition of DNAJC7 the rate of aggregation should follow a binding curve (ie more DNAJC7, more bound tau monomer, less aggregation). This seems to be the case, but the exact numbers do not match with the K_d of 0.5 nM – i.e. 0.1x of DNAJC7 should have very little effect on aggregation and 0.35x should show a ~20% decrease in rate. This problem is even more pronounced for P301L where the 1X curve is essentially the same as WT even though the K_d is 16x higher (see Fig1).

The CSPs saturate (and broaden) as we increase C7 concentrations. The concentration range of C7 is from 50 to 200uM and the concentration of ^{15}N tauRD used in the experiment is 100uM. In our experiments, the observed chemical shifts behave in accordance to the measured binding affinity. In the MST binding experiments the labeled tau is at much lower concentrations than in the ThT aggregation experiment which may explain differences between the two experiments given the affinity. It is also important to point out, that the ThT aggregation experiments are a relatively qualitative measure of aggregation and only detect the formation of beta-sheet structures but in reality the aggregation reactions contain a large complexity of structures/conformations that include monomers, seeds, oligomers and fibrils (and a dye). While we think we have measured affinities these only include binding between DnaJC7 and three possible species present in the starting sample: WT tau, P301L tau and WT tau monomer seed. thus at $t=0$ we have a good handle on the species distribution (tau monomer and seed) and how chaperone binding might influence the behavior but we do not know how DnaJC7 interacts with the other various intermediate species that are formed during the aggregation reaction. In principle, if the element that DnaJC7 recognizes in monomeric tau is available for interaction in tau assemblies it is possible that the chaperone can bind to oligomers and fibrils (assuming that the sterics of interaction are compatible). Furthermore, given the differences in tau fibril assemblies observed in the recently determined cryo-EM structures of tauopathy fibrils, it may be possible to predict what types of disease associated fibrillar assemblies DnaJC7 could bind. We are very interested in how molecular chaperones interact with different tau species to inhibit aggregation and have several ongoing projects in the lab to pursue these questions. Thus it is really challenging at this stage to extrapolate the effects of DnaJC7 on the complex mixture of species produced in the aggregation reaction.

Minor

1. Extreme care should be taken when modelling interactions between flexible proteins using sparse data. I am not sure what conclusions can be drawn from rosetta calculations using a homology model of DNAJC7 and a few crosslinks to dock tau. What does a representative model mean in such systems? Also, how do the calculations in Figure 2g-j show that R1R2 is more exposed?

This is a great point. We are really careful to only use modeling to produce large ensembles of conformations which we then cross validate using experimental restraints

derived from XL-MS experiments. This was our work flow for the R1R2 peptide docking experiments. Produce ensemble of R1R2 conformations that sample both “open” and “closed” conformations. In our prior work (Chen et al Nat Comm 2019) we showed that in tauRD each of these PGGG containing elements have some frequency in the ensemble. We thus used clustering to identify 20 representative conformations for the R1R2 region – this set of conformations was used then used to produce an unbiased ensemble of DnaJC7 docked with the R1R2 conformations, first using a low resolution centroid mode followed by a high resolution full atom mode. Each model was then scored to calculate a predicted binding energy by comparing the energetics of the unbound and bound states – At this stage production of the docked complexes is unbiased. We then compared this ensemble of conformations with the experimental crosslinks determined from DnaJC7 and tau and identified a small set of low energy structures that are compatible with the geometries of the crosslinker. While I agree, this is simply a theoretical model but several important hypotheses can be derived from these simulations: 1) the lowest scoring model of DnaJC7:R1R2 the peptide consistent with the crosslinks is bound in a collapsed conformation and 2) The collapsed R1R2 conformation is consistent with the observed crosslinks for tau. The structural model proposed here was then tested experimentally. First, we show that DnaJC7 is unable to bind a tauRD P270S mutant while it can bind to WT tauRD. Second, WT R1R2 which adopts a hairpin conformation should bind to DnaJC7 and blocks its ability to bind tau and thus block its aggregation suppression activity while the Proline to Serine mutant of R1R2 should be more expanded and thus should not bind to DnaJC7 allowing to bind to tau and thus inhibit aggregation. We think this identified a minimal element on tau that can bind to DnaJC7. We are currently pursuing an x-ray crystallography approach to determine a structure of DnaJC7 bound to the R1R2 peptide. Interestingly, DnaJC7 is not soluble at high concentration but incubation of DnaJC7 with the R1R2 allows us to concentrate the sample to 20mg/mls and several crystallization screen trials yielded promising crystals which we have reproduced and are awaiting to screen for diffraction.

2. What kind of chemical shifts (N, HN, CA, CB) were used in cs-rosetta? Is there any information about propensity for a local R1R2 structure in these?

The tauRD NMR chemical shifts used in CS-ROSETTA modeling include all backbone residues (N, HN, CA, CB). CS-ROSETTA uses a peptide fragment library derived from talos predictions of backbone torsional space from the chemical shifts. Albeit this is not perfect we think this ensemble is at least experimentally guided.

3. Please add a figure to show the position of all available lysines on the sequence or structure

This is a great suggestion – the TPR models in original Figure 5 partially had shown this but showing specifically lysines (and not lysines and arginines) on the full DnaJC7 structure is a great idea. We have included a pymol image of DnaJC7 shown in surface representation highlighting lysine residues in supplementary Figure 2d. We have included a brief description of the lysine distribution the description of the results for Figure 2. This does highlight that TPR2b (14) and the JD (14) have the most lysines in

DnaJC7 but there are sufficient numbers of lysines on the surface of TPR1 (9) and TPR2a (11) to detect crosslinks to tau (which is also lysine rich). These data suggest that specific TPR domain binding to tau determines the crosslinks and is not solely dependent on the frequency of lysines. This is also consistent with the overall amino acid composition of the different domains highlighting that TPR2b is overall more basic and contains the most lysines and arginines of all the domains on DnaJC7. Finally, our DnaJC7_{TPR2b_mut} highlights that specific mutations in the TPR2b binding groove yield a 20-fold reduction in affinity and capacity to suppress tau aggregation.

4. I found the use of 'natively folded tau' notion a bit confusing since the authors probably refer to an element that has some propensity for structure rather than being natively folded.

We have clarified this point. We think there are a combination of conformational changes in tau in response to a pathogenic mutation (i.e. P301L) which are local but also modify long range contacts. We have clarified this statement throughout the manuscript.

5. The authors propose that P301L changes the dynamics of the repeat domain using cross links at different temperatures. This is an intriguing experiment but I find it hard to interpret it as a change in dynamics.

We agree with the review that the term "dynamics" is not the right appropriate for what the XL-MS is capturing. We have rephrased "dynamics" to "conformation" or "alters crosslinking patterns" depending on the context.

6. 1.5 uM is mentioned as 0.35X in the caption but 0.5X in Figure 4 legend

We have corrected this typo in the caption.

7. Y Axis label is missing in Figure 5b

We have corrected the missing Y-axis in figure 5b (now Fig. 5d in revised manuscript).

We would like to end by thanking the reviewers for their obviously deep and thoughtful reading of our original manuscript. We hope that the revised text, including various new experimental results, addresses all of the reviewers' concerns and renders the paper ready for publication. In our view, the paper is much improved and carries a clearer and more focused message that will be of interest to many readers of Nature Communications.

References

1. Jerabek-Willemsen, M., et al., *MicroScale Thermophoresis: Interaction analysis and beyond*. Journal of Molecular Structure, 2014. **1077**: p. 101-113.
2. Yamaguchi, T., K. Matsuzaki, and M. Hoshino, *Transient formation of intermediate conformational states of amyloid-beta peptide revealed by heteronuclear magnetic resonance spectroscopy*. FEBS Lett, 2011. **585**(7): p. 1097-102.

REVIEWERS' COMMENTS

Reviewer #1 (Remarks to the Author):

The authors have included substantial new findings (stoichiometry, clearer descriptions of experimental methods, etc), which give greater confidence in the conclusions.

Reviewer #2 (Remarks to the Author):

The authors did a great job addressing the raised issues. I have no further comments. I strongly recommend publication of this interesting work

Reviewer #3 (Remarks to the Author):

The authors have addressed my previous concerns therefore I support the publication of this manuscript.

Reviewer #4 (Remarks to the Author):

The authors have made efforts to address one of my main concerns regarding the stoichiometry of the chaperone-substrate binding. Although I am not an expert in MST assays, the argument of comparing T-Jump and T-thermophoresis to extract stoichiometries seems rather weak. The presence of only a heterodimer after cross-linking is perhaps more compelling. Regarding the computational strategy employed, I do agree with the authors that their workflow is well thought out and tuned to provide a low-resolution description of a very dynamic system for which experimental data are sparse. However, the title talks about 'natively folded structural elements' and ' β -turn' is mentioned twice in the abstract. I failed to see any actual experimental evidence to support these statements in the manuscript and therefore I don't see how these important statements regarding the structural interpretation of the models produced with this elegant computational approach can be justified. Most importantly, I still have concerns about whether tau monomer sequestration is the mechanism that leads to inhibition of amyloid formation as the manuscript implies. For a high impact journal such as Nature Communications I would have expected to see a unified model that describes MST and aggregation assays in a quantitative manner, without discrepancies between experiments.

We are very grateful to the reviewers for their critical and detailed evaluation of our study which has resulted in a much stronger paper. We are excited that the reviewers overall agree that our revised work is suitable for publication in Nat Comm!

Reviewer #1 (Remarks to the Author):

The authors have included substantial new findings (stoichiometry, clearer descriptions of experimental methods, etc), which give greater confidence in the conclusions.

We thank the reviewer for taking the time to read our revised manuscript. We agree that the additional experiments have helped resolve some questions that were unanswered in the original submission.

Reviewer #2 (Remarks to the Author):

The authors did a great job addressing the raised issues. I have no further comments. I strongly recommend publication of this interesting work

We thank the reviewer for the overall positive feedback on the revised manuscript and that they strongly recommend our publication to be published in Nat Comm.

Reviewer #3 (Remarks to the Author):

The authors have addressed my previous concerns therefore I support the publication of this manuscript.

We thank the reviewer for the overall positive feedback on the revised manuscript.

Reviewer #4 (Remarks to the Author):

The authors have made efforts to address one of my main concerns regarding the stoichiometry of the chaperone-substrate binding. Although I am not an expert in MST assays, the argument of comparing T-Jump and T-thermophoresis to extract stoichiometries seems rather weak. The presence of only a heterodimer after cross-linking is perhaps more compelling. Regarding the computational strategy employed, I do agree with the authors that their workflow is well thought out and tuned to provide a low-resolution description of a very dynamic system for which experimental data are sparse. However, the title talks about 'natively folded structural elements' and ' β -turn' is mentioned twice in the abstract. I failed to see any actual experimental evidence to support these statements in the manuscript and therefore I don't see how these important statements regarding

the structural interpretation of the models produced with this elegant computational approach can be justified. Most importantly, I still have concerns about whether tau monomer sequestration is the mechanism that leads to inhibition of amyloid formation as the manuscript implies. For a high impact journal such as Nature Communications I would have expected to see a unified model that describes MST and aggregation assays in a quantitative manner, without discrepancies between experiments.

We thank the reviewer for taking the time to read our revised manuscript. We have addressed the final comments from the reviewer by rephrasing the description “natively folded” and “beta-turn” in the abstract. While we only have modeling and crosslinking experiments to support the concept that DnaJC7 recognizes these local structural elements, this work builds on our previous manuscript (Chen et al 2019), which implicated these inter-repeat domain local structures in tau as conformations that dictate the capacity of tau to aggregate or not. Finally, in regards to the mechanism of DnaJC7 recognition of tau binding and whether it involves natively folded species or seeds. Our binding data between DnaJC7:tau vs DnaJC7:monomer tau seed or DnaJC7:P301L tau clearly indicate a preference for binding aggregation-resistant forms of tau rather than aggregation-prone forms of tau. These are the primary data that we used to draw our conclusions for the novelty of this chaperone function. We are currently in the process of collecting high-resolution x-ray and cryo-EM data on DnaJC7 bound tau fragments or tau repeat domain to gain higher-resolution insight into this chaperones conformational selectivity. These high-resolution experiments will be essential to explain how DnaJC7 binds to different conformations of tau and not be confounded by the complexity of the tau conformational mixtures present in aggregation reactions.